



# Landcover succession for recently drained lakes in permafrost on the Yamal peninsula, Western Siberia

Clemens von Baeckmann[1,2], Annett Bartsch[1,2], Helena Bergstedt[1,2], Aleksandra Efimova[1,2], Barbara Widhalm[1,2], Dorothee Ehrich[3], Timo Kumpula[4], Alexander Sokolov[5], and Svetlana Abdulmanova[5]

[1]b.geos, Industriestrasse 1, 2100 Korneuburg, Austria
[2]Austrian Polar Research Institute, c/o Universität Wien, Austria
[3]UiT, The Arctic University of Norway, Tromso, Norway
[4]University of Eastern Finland, Joensuu, Finland
[5]Academy of Sciences Ural Branch of RAS to Arctic Research Station, IPAE UB RAS (Institute of Plant and Animal Ecology, Ural Branch, Russian Academy of Sciences)

**Correspondence:** Clemens von Baeckmann (clemens.von.baeckmann@bgeos.com)

**Abstract.** Drained Lake Basins (DLBs) are dominant features in lowland permafrost landscapes of the Arctic. Here we present a novel approach describing and quantifying the succession progression of recently drained basins using a landcover unit retrieval scheme developed specifically for the Arctic tundra biome. The added value compared to commonly used Normalized Difference Vegetation Index (NDVI) trend analyses is demonstrated. Landcover units were linked to DLB ages (years passed

since a drainage event occurred). The data were divided into bioclimatic subzones and the landcover units grouped according to their characteristics, first related to vegetation and second to wetness gradients (dry, moist and wet). A regression analyses of NDVI values and fraction of each landcover unit group provided the justification for the utility of the units in our research. The regression results showed the highest correlation with NDVI values for the wetness group 'Moist' and the vegetation group 'Shrub Tundra' ($R^2 = 0.458$ and $R^2 = 0.444$). There was no correlation ($R^2 = 0.066$) found between NDVI and the

fraction of group 'Wet'. This highlights the importance of an alternative to NDVI such as the use of landcover units to describe wetland area changes. Finally, our results showed different trajectories in the succession of landcover units in recently DLBs with respect to different bioclimatic subzones. Remaining water in the basin after a lake drainage event was highest for the most southern subzone (median 6.28 %). The open water fraction dropped below one percent for all subzones after five to ten years since drainage. The results of this study contribute to an improved understanding of DLB landcover change in permafrost

environments and to a better knowledge base of these unique and critically important landforms.

## 1 Introduction

Arctic permafrost regions are undergoing drastic changes due to climate change, leading to widespread, unprecedented landscape disturbances and changes (IPCC, 2022). Those are both of gradual and abrupt nature and range from plot scale to large scale, a regional to circumpolar phenomenon (Turetsky et al., 2020). Increasing temperatures induce the thawing of ice-rich

permafrost or the melting of ice and greenhouse gases, methane and carbon dioxide are released into the atmosphere (Turet-



sky et al., 2020; Manasypov et al., 2022; Schuur et al., 2022). The forming of characteristic landforms can be described as a process of the disturbance of the thermal equilibrium of the ground (Jones et al., 2011). Closed depressions filled with water, formed by the settlement of the ground caused by the thermo-induced process, are called thermokarst lakes. Formation and drainage cycles span over thousands of years. Those are common features covering 50% to 75% of permafrost lowlands in parts of Alaska, Siberia, and Canada (Jones et al., 2022). The distribution of thermokarst lakes, DLBs and the associated cycles (formation and drainage) differ between areas of different ground-ice content and between regions (Subarctic, Arctic and high-Arctic). The resulting mosaic provides a unique habitat for both flora and fauna. Arctic landscapes are widely influenced by DLBs and their role for the geomorphological, hydrological and ecological development in these regions (Jorgenson and Shur, 2007). The drainage of thermokarst lakes is a long-existing phenomenon in many lowland permafrost regions prone to thermokarst landforms, with DLBs of several thousand years of age present among existing lakes and recently drained basins. Studies suggest that the frequency of drainage events has been increasing with climate change recently (Smith et al., 2005; Carroll et al., 2011; Kanevskiy et al., 2013; Nitze, 2018; Nitze et al., 2020; Arp et al., 2023; Chen et al., 2023). Lake changes and drainage processes can be observed using remote sensing methods. For example: the change in water area due to shrinking, expanding and disappearing of the surface water has been observed using a multitude of different satellite based sensors as multiple studies showed (Grosse et al., 2013; Karlsson et al., 2014; Nitze et al., 2020; Bergstedt et al., 2021; Jones et al., 2022; Bartsch et al., 2023b). Monitoring lake drainage requires separation of open water from land area. The detection of previously drained lakes requires spatial data analyses in order to identify relevant depressions and gain knowledge about trajectories of landsurface hydrology change and vegetation succession is needed. The applicability of remote sensing data for identifying DLBs has been demonstrated (Smith et al., 2005; Jones et al., 2011; Bergstedt et al., 2021).

Landcover in lake basins changes due to the decrease in open water area and the establishment of vegetation communities. Drained lakes basins become sites of plant communities with high biomass and productivity, peat accumulation, permafrost aggradation and critical areas for the carbon cycle (Grosse et al., 2013; Göckede et al., 2019; Loiko et al., 2020; Bergstedt et al., 2021; Jones et al., 2022; Manasypov et al., 2022). Previous studies have focused on the usage of the Normalized Difference Vegetation Index (NDVI). Chen et al. (2021) and Liu et al. (2023) identified the largest changes within the first five years after drainage for the NDVI. A spatial analysis of a multi-basin data set of landcover succession in recently drained basins has not been addressed yet. Mapping landcover and vegetation communities in the Arctic requires specialised approaches (Bartsch et al., 2016). Methodologies must be able to monitor the unique characteristics of Arctic landscapes and vegetation communities. Landcover classifications in permafrost regions are used especially for upscaling of soil properties and fluxes (Bartsch et al., 2019b). The Circumpolar Arctic Vegetation Map (CAVM, (Walker et al., 2005)) provides detailed information for vegetation communities but at comparably coarse spatial detail (1 km). Bartsch et al. (2023a) presented a novel approach of landcover characterization (10 m nominal resolution), with representation of moisture gradients and vegetation physiognomy making it uniquely suitable for the analysis of DLBs. Actual landcover information can be potentially used for flux upscaling and studies that focus on habitat changes. The approach depends, however, on the fusion of Sentinel-1 and Sentinel-2 data, both Copernicus satellite missions, with availability limited to 2016 onwards.





Several regions with frequent lake drainage have been identified previously. For example, Nitze (2018) investigated regions in Alaska, Eastern and Western Siberia. The latter includes the Yamal peninsula which is subject to a multitude of different environmental changes over the past 50 years, caused both by climate change as well as direct impact of anthropogenic activity (Kumpula et al., 2010, 2011, 2012). Abundant lakes and DLBs in parts of Yamal create terrain which is important both as habitat and unique grazing ground for the local reindeer herding community. Changes in the landscape influence those habitat
and affect the local biodiversity (Kumpula et al., 2011, 2012).

The aim of this study was to investigate the impacts of lake drainage on landcover change along a bioclimatic gradient on the Yamal peninsula, specifically focusing on the change of wetness gradients and vegetation composition.

As vegetation growth is climate dependent, different bioclimatic subzones were considered. While Bartsch et al. (2023a) focused on a static landcover product based on Sentinel-1/2 with a high consistency on circumpolar scale we focused on time
series of selected DLBs for 2016 onwards. To tackle the issue of data availability in the years before, a space-for-time concept was applied. This requires additional knowledge on the drainage date. Pre- 2016 drainage ages were determined using Landsat products (30 m) dating back to 1984.

## 2   Study area and data

### 2.1   The Yamal peninsula

The Yamal peninsula (Western Siberia) has been subject to a multitude of different environmental changes over the past and represents multiple bioclimatic subzones (Figure 1). The peninsula is north- south orientated with a distance of 780 km covering a biogeographical gradient from the forest tundra to the high Arctic. Yamal is located in the north-western part of the West Siberian Plain, the highest peak on the peninsula is about 80 m elevation (Kumpula et al., 2012). Discontinuous and continuous permafrost is present in this region (Figure 1). The permafrost fraction is increasing to the north on the Yamal peninsula (Obu
et al., 2021). Many landsurface changes are associated with permafrost warming in this transition zone (Babkina et al., 2019). Yamal is covered by different tundra vegetation communities, thaw lakes, wetlands and river floodplains. The peninsula has the second highest lake and drained lake basin coverage (78%), following only the Yukon–Kuskokwim Delta (Jones et al., 2022). Disappearing lakes were previously reported specifically for the southern part (Smith et al., 2005; Nitze, 2018). Central Yamal is known for rising temperatures and changes associated with unusually warm summers (thaw slumps, active layer deepening
etc.; e.g. Babkina et al., 2019; Bartsch et al., 2019a). The whole region has been shown to be a hot spot of thaw lake change (Nitze, 2018). Drained thaw lake basins differ from the surrounding areas in vegetation composition and permafrost conditions, introducing additional heterogeneity into the tundra landscape (Jones et al., 2022) impacting habitat availability for both flora and fauna. Drained thaw lake basins differ across the regions of Yamal in their size and abundance. Wetlands are the dominant landscape type throughout the region. Low erect shrub willow (Salix) is abundant on Yamal and covers valleys and slopes
(Kumpula et al., 2012; Leibman et al., 2015). In central Yamal, in bioclimatic subzone D, the Vaskiny Dachi research station is located and extensive slope process activation resulting in cryogenic landslides are monitored since 1989 (Leibman et al., 2015).




One of the world's largest gas deposits is located in this region of Yamal. Human activity does not only comprise gas infras-
tructure projects but extensive reindeer herding serves as main traditional land use form. Nomadic reindeer herders, the Nenets
move annually between winter pastures at the tree line and the northern tundra which exposes them to impacts associated with
exploration and production activities (Kumpula et al., 2010; Volkovitskiy and Terekhina, 2023; Spiegel et al., 2023).

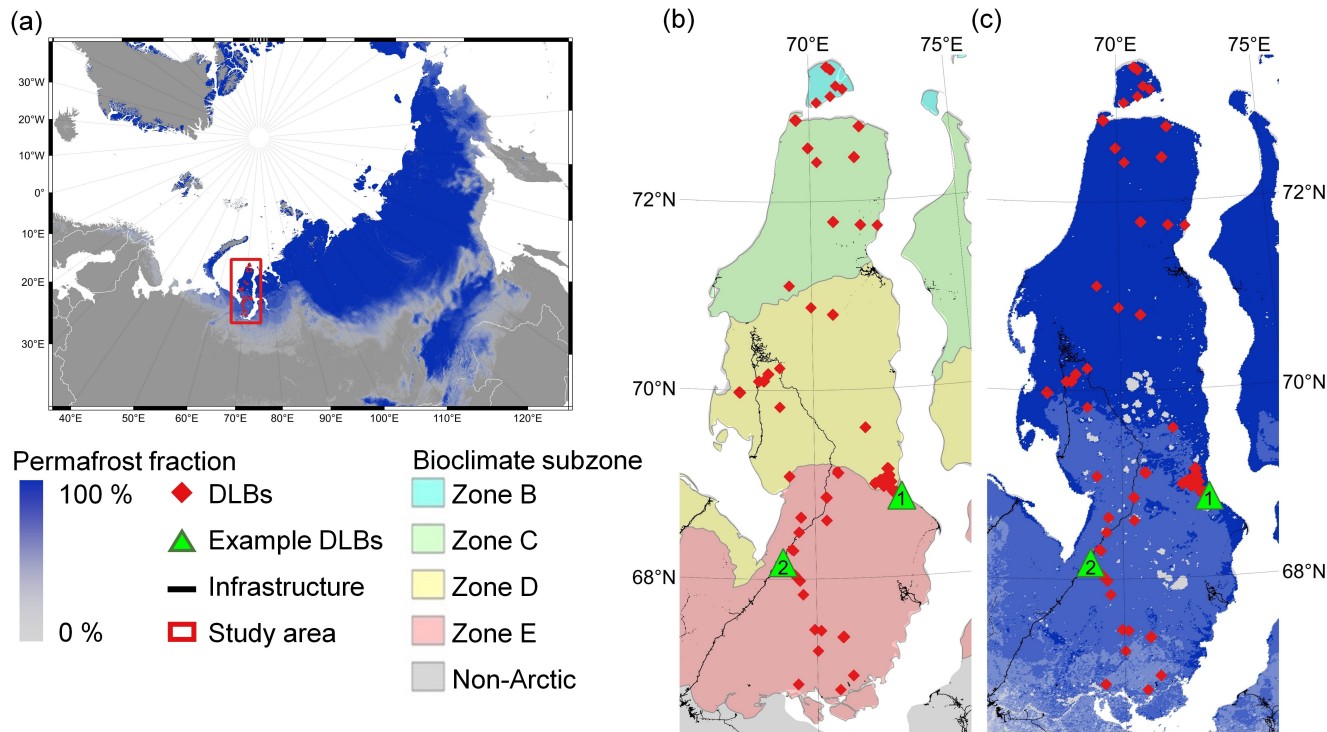

**Figure 1.** Study area characteristics: a) location of the study site (red rectangle), b) bioclimate subzones based on (Walker et al., 2005) with
subzones *B) Arctic tundra- northern variant, C) Arctic tundra- southern variant, D) Northern hypo-Arctic tundra, E) Southern hypo-Arctic
tundra*, and c) permafrost fraction (Obu et al., 2021). Infrastructure information (Bartsch et al., 2023c) is shown in black. Green triangles
with the numbers (1) and (2) represent locations of the examples introduced in this study (1 - DLB time series, 2 - *in situ* data).

Monthly mean temperatures of the bioclimatic subzones reflect the gradient (Figure A1). Precipitation is on average around
300 mm annually. In winter snow accumulates up to 30 cm thick on flat surfaces, in some areas on the leeward slopes and in
the valleys it can accumulate up to several meters (6 m) (Leibman et al., 2015).

95  **2.2  Data**

The presented study builds on data from the Sentinel missions derived *via* the Europe's Copernicus programme run by the
European Commission (EC) and the European Space Agency (ESA). We used the radar and the multispectral high-resolution
acquisitions from Sentinel-1 and Sentinel-2.



### 2.2.1 Sentinel data

Sentinel-1 is a radar imaging mission using C-band SAR and is operating during day and night (ESA, 2012). There are two operational, polar-orbiting satellites which use an identical C-band SAR sensor (Schubert et al., 2017):

- Sentinel-1A, launched on 3[rd] April 2014 and

- Sentinel-1B, launched on 25[th] April 2016 - mission ended in 2021.

The satellites provide all-weather imagery of earth's surface. They orbit 180° apart allowing for a six-day revisiting period. However, the revisiting period varies and is only 12 days for most Arctic regions. For this study we used data acquired in VV polarization (Vertically emitted and Vertically received) which is most sensitive to rough surface scattering caused from water or bare soil. Data availability in recent years is, however, limited for the Yamal region due to the failure of Sentinel-1B in 2021.

Sentinel-2 is a multispectral high-resolution imaging mission and is monitoring on 13 different spectral bands. There are two operational, polar-orbiting satellites (ESA, 2015):

- Sentinel-2A, launched on 23[th] June 2015 and

- Sentinel-2B, launched on 7[th] March 2017.

Both satellites are identical and operate in the same orbit. They provide a resolution up to 10 m (depending on the spectral band), and achieve a five day revisit period when used in combination.

### 2.2.2 Auxiliary data

Daily temperature data are required for Sentinel-1 scene selection. They are derived using the ERA5-Land dataset product from the Copernicus Climate Change Service (C3S) at the European Centre for Medium-Range Weather Forecasts (ECMWF). This fifth generation of atmospheric reanalysis data for global climate parameters covers a period from 1940 up to present. It combines model data with observation data merged into a global dataset which is consistent over time (Hersbach et al., 2020). The data were downloaded *via* the Climate Data Store (CDS) portal. Air temperature at two meters above the surface were used for the selection of Sentinel-1 scene observations from the "ERA5 hourly data on single levels from 1940 to present" dataset. The spatial resolution is a 30 x 30 km grid in the horizontal (Hersbach et al., 2020).

Pre-drainage lake extent is derived using the Landsat product published by Nitze et al. (2018) adopted from the Hot Spot Regions of Permafrost Change ("Hot Spot Product") by Nitze (2018). This trend dataset is generated using the spectral indices: Tasseled Cap (TC) Brightness, Greenness, and Wetness; NDVI; Normalized Difference Moisture Index (NDMI); and Normalized Difference Water Index (NDWI). The indices provide information on surface properties, such as albedo, vegetation or moisture/water on the earth's surface (Nitze, 2018). The main data source for the calculations is taken from the Landsat time-series stacks supplied by the United States Geological Survey (USGS). Data between the years 1999 and 2014 were used



to calculate the slope and intercept parameters *via* the robust Theil-Sen regression algorithm. Only data for the peak summer
season (July, August) was taken from the Landsat achieve. The resolution of the dataset is 30 meter. The final product extents
over several regions in the Arctic: Western Siberia, Eastern Siberia, Alaska and Eastern Canada. The study region Western
Siberia covers the whole Yamal Peninsula and is used for our study. Data are available as geospatial vector files for 218 882
individual lakes. For each of the lakes certain attributes were given. This included: Area at start of observation (1999) in ha,
Area at end observation (2014) in ha, Net lake area change in ha, Net lake area change in percent *etc.*

Bioclimatic zones were obtained from the Circumpolar Arctic Vegetation Map (CAVM), (Walker et al., 2005) which was
developed by vegetation experts. It offers plant community information north of the treeline and distinguishes between 16
classes. This widely used landcover product provides additional information including the dataset for different bioclimatic
subzones in the Arctic regions. This sub- dataset on bioclimate subzones separates the Arctic in five different zones. The
boundaries are based on phytogeographic subzones (the geographic distribution of plant species) based on recent information
from numerous sources (Raynolds et al., 2019). On a gradient from north to south, there is an increase in corresponding plant
size, coverage, abundance, productivity, and species richness. In the coldest parts fewer then 50 species which form plant
communities regarding their number of plants in local floras are available. They increase to more then 500 species in the
southern zones near the treeline (Raynolds et al., 2019):

- Subzone B, 'Arctic tundra- northern variant', mean July temperatures are about 3 - 5 °C, woody plants first occur as
  prostrate (creeping) dwarf shrubs, and increase in stature to hemiprostrate dwarf shrubs (<15 cm tall),

- Subzone C, 'Arctic tundra- southern variant', mean July temperatures reach about 5 - 7 °C, erect dwarf shrubs (<40 cm
  tall) and woody plants increase in stature to hemiprostrate dwarf shrubs (<15 cm tall),

- Subzone D, 'Northern hypo-Arctic tundra', mean July temperatures are about 7 - 9 °C, low shrubs (40-200 cm tall) and
  erect dwarf shrubs (<40 cm tall) are present, and

- Subzone E, 'Southern hypo-Arctic tundra', mean July temperature range about 9 - 12 °C, low shrubs (40-200 cm tall)
  are present. Woody shrubs are abundant and reach up to 2 meters in height closer to the treeline, where the mean July
  temperatures are between 10 and 12 °C.

**3 Methods**

In a first step, a database of recently drained lakes was created based on the Landsat trend product (Nitze et al., 2018). The
trends were used for identification of potentially drained lakes before the availability of Sentinel-1 and Sentinel-2, between
1999 and 2014, to facilitate space for time assessment. The basin ages were determined using the change of the water area and
the corresponding landcover year. In a second step cloud free Sentinel-2 images were used to identify drainage events from
160  2016 onwards and the calculation for the corresponding basin ages for events after 2015 (basin ages younger then 2015) until



the year 2022. Landcover was derived using Sentinel-1 and Sentinel-2 data. The landcover units were grouped by vegetation type as well as wetness gradients. *In situ* data collected within DLBs was utilized to justify the grouping of the wetness gradients. NDVI was obtained from Sentinel-2 in addition. In a final step, the landcover and NDVI data were combined with the drainage age database covering the years before 2015 to enable space for time assessment.

## 3.1 Lake selection and age determination

A first selection of potential DLB objects was carried out using the Landsat product by Nitze (2018). Lakes with at least 50% of drainage were considered in the following processing. The main focus was set on the lake position to cover all present bioclimate subzones on the Yamal peninsula. A second filtering was done *via* visual inspection to exclude false detected DLBs caused by e.g. flood plain related water area change. Lakes were also excluded when no drainage age could be detected. This is likely due to refilling, unclear drainage or when the lake drained before 1984. The determination of the drainage year and the removing of the falsely detected DLBs were done by visual inspection of the actual Landsat acquisitions. Google Earth Pro Historical Landsat Timelapse Imagery data were used for each selected object on a timeline starting from 1984. A different method was used for lakes which drained between 2015 and 2022. The landcover unit 'Water' (#1, as defined in Bartsch et al. (2023a)) was investigated based on annual classifications of Sentinel-1 and -2. Focus was set on areas that are not classified as 'Water' for the following year. Inspection of the original Sentinel-2 acquisitions was performed and the basin excluded when the described conditions occurred. The Landsat dataset provided the corresponding maximum lake extend (water area extent in 1999) for consistency and the DLB was included into our database. The DLB age is the difference between the year of the used landcover and the year when the drainage event took place. Only DLBs with a maximum water extent (status before drainage) of larger than 0.01 km$^2$ were considered.

## 3.2 Sentinel-1/2 pre-processing

### 3.2.1 Sentinel-1

The Sentinel-1 Interferometric Wide (IW) swath mode data were downloaded from the Copernicus open access hub. Only vertically transmitted - vertically received (VV) polarisation information was used following the scheme of Bartsch et al. (2023a). The effect of temperature on the backscatter data (Bergstedt et al., 2018; Bartsch et al., 2023b) was minimized by using acquisitions only within a certain ground temperature range derived using reanalyses data (ERA5). The defined temperature window ranged from -10 to 0 °C. To exclude the influence of soil moisture dynamics on the backscatter during unfrozen conditions, temperature below freezing point was used only. The minimum temperature was defined at -10 °C because colder temperature would result in higher backscatter as expected (Bergstedt et al., 2018). The Sentinel-1 processing was done using the SNAP toolbox provided by ESA. The steps included: applying precise orbit information, thermal noise removal, calibration, orthorectification using the Copernicus digital elevation model (DEM) at 90 m resolution, conversion to dB values and normalisation (Widhalm et al., 2018). The Sentinel-1 data acquired during frozen conditions represent surface roughness in tundra regions and complements the information contained in the Sentinel-2 bands.



### 3.2.2  Sentinel-2

The Sentinel-2 data were downloaded as Top of Atmosphere (TOA) Level-1C product and processed to Bottom of Atmosphere
(BOA) Level-2A product. Atmospheric-, terrain- and cirrus corrections were performed using the Sen2Core processor toolbox
from ESA. We then processed the data with a super-resolution approach using Dsen2, (Lanaras et al., 2018) for the Sentinel-2
bands to utilize their multi-spectral capabilities due to the different spatial resolution (bands B03, B04 and B08 at 10 m and
B11 and B12 at 20 m). The tool Dsen2 (Lanaras et al., 2018) outperformed simpler upsampling methods because it preserved
spectral characteristics (Bartsch et al., 2023a). The main difference to other upscaling methods was that a convolutional neural
network was used. Clouds were masked using the Scene Classification Map (SCM) from the Sen2Core product on the upscaled
version.

### 3.3  NDVI, landcover and regression calculation

The NDVI was calculated on the BOA values. The change in NDVI values were associated with the loss of water area (negative
NDVI values) and colonization of the vegetation (positive NDVI values) in the basin. The yearly NDVI data were masked for
open water bodies using the corresponding landcover product for each year during the investigated time period (2015 - 2022).

A supervised maximum likelihood classification was performed on the processed Sentinel-1 and Sentinel-2 data. Five spec-
tral bands were utilized from the Sentinel-2 satellite data: Band 3 (green) at 10 meter spatial resolution, 4 (red) at 10 m, 8
near-infrared (NIR) at 10 m, 11 shortwave-infrared (SWIR) at 20 m and band 12 (SWIR) at 20 m. Acquisitions between July
and August were selected to cover the mid-growing season for vegetation. The units for the supervised classification were
originally derived from a k-means unsupervised classification done by Bartsch et al. (2019b) on a 100 km wide and 1400 km
long transect in Western Siberia (along the 70° meridian, from 61° North to 74° North). For this study, we used the updated
scheme from Bartsch et al. (2023a), also referred to as CALU (circumarctic Landcover Units).
In a final step the separation into the different landcover units from the CALU product was modified (Table 1). 'Shadow' was
often missclassified over lakes due to lower values in the spectral bands. The landcover unit 'Water' (#1) was therefore merged
with 'Other, shadow' (#23) into one 'Water' unit #1. The original CALU approach used three different input images from
Sentinel-2 for robustness of the retrieval (e.g. handling potential haze, failure of cloud masking). Only one input image from
Sentinel-2 was used to keep the temporal information on each year in our approach. The naming, coloring and the grouping for
the landcover units followed Bartsch et al. (2023a). Two groups were introduced combining vegetation units and to distinguish
between wetness gradients (Table 1):

- **Group A:** Seven different units focusing on vegetation assessment considering shrub types and



– **Group B:** Four different wetness gradients according to their wet- dry conditions. The forest units and the #1 unit 'Water' were also included in addition to the original grouping. The 'Recently burned or flooded, partially barren' landcover unit #17 was not considered.

The wetness gradients (Group B) were assessed with *in situ* data to justify the differentiation in this study.

**Table 1.** Legend for the land cover units based on (Bartsch et al., 2023a) and grouping schemes.

| ID | Description | Group A | Group B |
|----|-------------|---------|---------|
| 1 | water | water | water |
| 2 | shallow water / abundant macrophytes | wetland | wet |
| 3 | wetland, permanent | wetland | wet |
| 4 | wetland, seasonally inundated | wetland | wet |
| 5 | moist tundra, abundant moss, prostrate shrubs | grassland | moist |
| 6 | dry to moist tundra, partially barren, prostrate shrubs | lichen and moss | dry |
| 7 | dry tundra, abundant lichen, prostrate shrubs | lichen and moss | dry |
| 8 | dry to aquatic tundra, dwarf shrubs | shrub tundra | moist |
| 9 | dry to moist tundra, prostrate to low shrubs | shrub tundra | moist |
| 10 | moist tundra, abundant moss, prostrate to low shrubs | shrub tundra | moist |
| 11 | moist tundra, abundant moss, dwarf and low shrubs | shrub tundra | moist |
| 12 | dry to moist tundra, dense dwarf and low shrubs | shrub tundra | moist |
| 13 | moist tundra, dense dwarf and low shrubs | shrub tundra | moist |
| 14 | moist tundra, low shrubs | shrub tundra | moist |
| 15 | dry to moist tundra, partially barren | shrub tundra | moist |
| 16 | moist tundra, abundant forbs, dwarf to tall shrubs | shrub tundra | moist |
| 17 | recently burned or flooded, partially barren | shrub tundra | |
| 18 | forest (mixed) | forest | moist |
| 19 | partially barren | barren | dry |

The basins were analyzed separately with respect to the bioclimate subzones. A buffer area of 1 km extent was derived based on the maximum water extent and was referred to "peripheral area". A regression analysis was carried out for the NDVI values (dependent variable) and the fraction for the units of group A and B (independent variable).

### 3.4 *In situ* data collection

*In situ* vegetation information has been collected in a drained lake basin on July 26[th] in 2016 (for location see Figure 1). Vegetation cover was recorded in eight 2 x 2 m sampling plots arranged along a transect extending from the edge of the old DLB to the open water (Figure A2). The plots were chosen to represent different vegetation assemblages. Coverage was recorded visually for the most important plant groups (e.g. forbs, shrubs) or genera (*Salix, Eriophorum*; see table A1 for all recorded categories). The vegetation of the drained area was also documented with pictures. The featured lake had water level fluctuations since at least 1984. The recent basin was formed during a drainage event around 2010. The soil of the basin is clay. Some litter was found in the sample areas as well. Total vegetation cover was 80% or above at the date of the fieldwork.





## 4 Results

### 4.1 Basins and age

In total, 51 different recently DLBs were identified and analyzed. The selected DLBs events ranged from 1998 (one event) till 2021 (also one event). The most drainage events were detected in 2019 (15 events). 13 drainage events were included into the database for occurrences before 2015, and 28 events after. The open water extend (before drainage) ranged between 0.02 km$^2$ and 6.88 km$^2$. The number of available lakes strongly varied for each of the bioclimate subzone. Six recently DLBs were selected in bioclimate subzone B (0.55 km$^2$ of total drained area, average: 0.09 km$^2$), which expands over Bely Island north of

the Yamal peninsula. Heading southwards, also six recently DLBs were selected for bioclimate subzone C (3.41 km$^2$ drained area, average: 0.57 km$^2$). 17 DLBs were selected for bioclimate subzone D (27.85 km$^2$ drained area, average: 1.64 km$^2$) and 22 for subzone E (19.68 km$^2$ drained area, average: 0.89 km$^2$). Drainage event information was collected for a total of 30 consecutive years for the entire study area. Basin ages in our database ranged from -5 to 24 years, where year 0 represents the year of the event of the drainage. Splitting the data into the bioclimate subzones resulted in age gaps and inconsistent time

series respectively. Further analyses were therefore limited to a basin age of 10 years. Age groups (1 - 5 and 6 - 10 years) were built to increase the number of recently DLBs. The water unit was analysed in detail and all units were compared to average fraction of their respective subzone.

### 4.2 NDVI change

NDVI magnitude and change over time differed across the bioclimatic zones (Figure 2). Within the first three to four years,

NDVI values inside the basin increased to the level of peripheral values for the southern subzones D and E. The subzones B and C did not show such an increase within this time. Subzones B and C showed higher median NDVI values (B: 0.34, C: 0.33) already in the year of the drainage event (DLBs age 0) compared to the southern subzones (D: 0.03 and E: 0.10) (Figure 2).

**Table 2.** Median NDVI (Normalized Difference Vegetation Index) values derived from Sentinel-2 for lake basins up to 10 years after drainage, separated by bioclimatic subzone (B to E, see Figure 1 ). No data is indicated with a line (-).

| | Bioclimate subzone | | | | | Bioclimate subzone | | | |
|---|---|---|---|---|---|---|---|---|---|
| year | B | C | D | E | year | B | C | D | E |
| 1 | 0.26 | 0.23 | 0.23 | 0.19 | 6 | 0.20 | 0.38 | 0.45 | 0.64 |
| 2 | 0.34 | 0.30 | 0.31 | 0.30 | 7 | 0.52 | 0.49 | 0.60 | 0.61 |
| 3 | 0.33 | 0.24 | 0.46 | 0.53 | 8 | - | 0.45 | 0.55 | 0.62 |
| 4 | 0.28 | 0.31 | 0.60 | 0.60 | 9 | - | 0.50 | 0.53 | 0.60 |
| 5 | 0.26 | 0.33 | 0.69 | 0.63 | 10 | - | 0.52 | 0.55 | 0.67 |





The NDVI for the peripheral area appeared stable considering the entire time period, although some year to year differences occurred (Figure 2). Peripheral median NDVI values increased on a north to south gradient from subzone B: 0.48, subzone C:

0.51, subzone D: 0.55, to subzone E: 0.58.

**Figure 2.** NDVI change within the first ten years after drainage separated by bioclimatic subzone (a - d). The peripheral area refers to the area around the DLBs for a zone of 1 km.

## 4.3   Landcover succession

Landcover succession differed across the subzones (Figure 3):

– **Bioclimate subzone B:** four years after the drainage event the 'Water' (#1) fraction dropped to ≤10%. The 'Shallow water' area (#2) decreased from ~20% to 11%. Permanent wetland (#3) dropped to ≤10% at the age of four but increased

265        to ~13% at the age of seven. 'Dry tundra with abundant lichen' (#6) increased in the seventh year to 48%.



- **Bioclimate subzone C:** 'Water' dropped to ≤10% after two years since drainage. The wetland type changed over time. #2 stayed ∼30% until year eight and then decreased to ∼2%, #3 increased to ∼25% until year six and then decreased again ≤10% in year 10, #4 stayed ≤10%, except year nine when the fraction increased to 17%. 'Partially barren, dry to moist tundra' (#15) decreased in the first two years from 34% to ≤20% and stayed ≤20% for the rest of the years, #16 stayed ≤1% for all years, except for the last two years when the fraction increased to ≤10%. 'Partially barren' (#19) increased to 34% within the initial five years, then decreased and stabilized at ∼20%.

- **Bioclimate subzone D:** the 'Water' fraction in the basin dropped to ≤10% after the basin reached an age of three years. The fraction of unit #2 decreased from 30% by the fifth year and then stayed ≤10%, #4 increased until the fourth year to ∼10% and then stayed until year seven at ∼10% but decreased after that to ≤5%. #15 stayed ∼25% until year seventh and then increased to 54% for the following two years, #16 increased until the seventh year to 33% and then decreased again, 'Partially barren' (#19) decreased from 28% until an age of five years and then stayed at ≤10%.

- **Bioclimate subzone E:** 'Water' decreased from 15% to ≤1% after the year eight. #2 stayed ∼20% for the first four years and then decreased to 4%, #4 increased to 20% in the seventh year and dropped in the following years to ≤10% again. #9 slowly increased to 10% for the 10th year, #10 stayed ≤5% until the fifth year and then increased to 15% for the ninth year, #11 slowly increased to 10% by the 10th year of the DLBs.

 

**Figure 3.** Landcover change in time for the DLBs separated by the different bioclimate subzones (a - d). The sum of available basins for a certain age is displayed on top. For color legend see Table 1.

An example for landcover change in a drained lake located in subzone E, is shown in Figure 4. The DLB is close to subzone D, on the east side of the Yamal peninsula (Figure 1) and is indicated as green triangle with the number 1 on top. The annual landcover showed a decrease (more then 90%) in the 'Water' landcover unit from 2016 to 2017 (Figure 4, a and b), when the drainage event took place. Small 'Wetlands' formed during the years after drainage especially in the southern part of the basin. The barren units #19 and #15 were present with a fraction of over 50% for 2017 and 2018 (Figure 4, b and c) after the 'Water' fraction decreased. For the year 2019 (Figure 4, d), wetlands formed ('Wetland, permanent wetland' and 'Wetland, seasonally inundated') which got mostly replaced in the following years (2020 - 2022, Figure 4, e - g) with #16 'Moist tundra, abundant forbs, dwarf to tall shrubs'. More then 50% is classified as CAL Unit #16 in 2022, the last year of the generated landcover.





**Figure 4.** Annual landcover for the example lake in zone E (2016 - 2022; a - g), for location see Figure 1. Fractional change of landcover units across the total lake area (h). For landcover unit legend see Table 1.

The grouping of yearly basin ages into two categories (1 - 5 years and 6 - 10 years) resulted in an increase in the number of basins within each specific bioclimate subzone. In the first age category, subzone B had 28 basins, subzone C had 18, subzone D had 53, and subzone E had 64. In the second age category, subzone B had 5 basins, subzone C had 14, subzone D had 19, and subzone E had 17.



**Table 3.** Median fraction of landcover units (see Table 1) for the two age groups (1-5 years and 6-10 years), separated by bioclimate subzones (B to E, see Figure 1). Values above 5% are highlighted in **bold**.

| | #1 | #2 | #3 | #4 | #5 | #6 | #7 | #8 | #9 | #10 | #11 | #12 | #13 | #14 | #15 | #16 | #17 | #18 | #19 |
|---|---|---|---|---|---|---|---|---|---|---|---|---|---|---|---|---|---|---|---|
| **B (1-5)** | 2.42 | **19.32** | **6.65** | 2.04 | 0.00 | 2.24 | 0.92 | 0.00 | 0.00 | 0.00 | 0.00 | 0.00 | 0.00 | 0.00 | 3.16 | 0.00 | 0.00 | 0.00 | **32.55** |
| **B (6-10)** | 0.13 | **11.25** | **13.98** | 1.68 | 0.00 | 3.97 | 4.94 | 0.00 | 0.00 | 0.34 | 0.00 | 0.00 | 0.00 | 0.00 | 0.67 | 0.00 | 0.00 | 0.00 | **12.33** |
| **C (1-5)** | 3.55 | **21.50** | 4.43 | 0.73 | 0.05 | 4.15 | 0.55 | 0.15 | 0.17 | 0.00 | 0.00 | 0.00 | 0.00 | 0.03 | **13.33** | 0.00 | 0.00 | 0.00 | **13.09** |
| **C (6-10)** | 0.02 | **15.90** | **17.84** | **5.97** | 0.02 | 3.81 | 0.50 | 4.45 | 0.41 | 0.13 | 0.08 | 0.20 | 0.02 | 0.57 | **12.52** | 0.38 | 0.00 | 0.34 | **6.51** |
| **D (1-5)** | 2.78 | **20.33** | **6.26** | 1.25 | 0.00 | 0.32 | 0.14 | 1.20 | 0.40 | 0.06 | 0.05 | 0.03 | 0.00 | 0.24 | **15.77** | 0.34 | 0.00 | 0.00 | **6.56** |
| **D (6-10)** | 0.86 | **6.22** | 4.91 | 3.57 | 0.20 | 2.45 | 3.57 | 4.86 | 4.10 | 1.61 | 0.67 | 0.07 | 0.01 | 1.18 | **25.95** | **15.04** | 0.00 | 0.44 | 1.90 |
| **E (1-5)** | **6.28** | **19.80** | 4.65 | 1.99 | 0.00 | 0.51 | 0.14 | 1.52 | 0.15 | 0.05 | 0.05 | 0.00 | 0.00 | 0.13 | **18.24** | 0.53 | 0.00 | 0.00 | **10.90** |
| **E (6-10)** | 0.00 | 3.03 | **9.03** | **8.80** | 1.06 | 0.71 | 0.59 | 3.99 | 5.93 | **9.18** | 2.91 | 0.21 | 0.06 | 0.87 | **5.62** | **6.78** | 0.00 | 0.49 | 0.56 |

In all subzones beside the subzone B, more shrub tundra units (#8 - #17) were present in the second age category (6 - 10 years) compared to the first group (Table 3). Additionally, a more heterogeneous distribution of the landcover units was observed for the southern subzones D and E compared to the northern zones. A higher proportion of the wet units (#2 - #4) was shown in all subzones, with the exception of subzone B, where unit #4 was still below average.





**Figure 5.** Landcover distribution for two age categories: left (a, c, e, g) 1 - 5 years, right (b, d, f, h) 6 - 10 years after drainage. Different bioclimate subzones on Yamal are addressed. The number of available DLBs is shown on the top right of each sub plot. For landcover unit legend see Table 1. Subzones are shown in Figure 1. Grey bars represent the average landcover distribution for each of the bioclimate subzone.





The fraction of the 'Water' unit (#1) remains larger then 0% after a drainage event (Table 4). The remaining 'Water' fraction for the first age category (1 - 5 years) was the highest for bioclimate subzone E with 6.28 % and the lowest in the median for the second age category (6 - 10 years). 'Water' fraction dropped below 1% for all subzones in the second age category. The values in Table 4 include also 0% data to retain the full temporal information regarding the 'Water' unit (#1).

**Table 4.** 'Water' fraction [%] for DLBs separated into the two age groups and the bioclimate subzones. The fraction is given for the quantiles: 25, 50 (median) and the 75.

| | 1-5 [years] | | | 6-10 [years] | | |
|---|---|---|---|---|---|---|
| **Bioclimate subzone** | **q25** | **q50** | **q75** | **q25** | **q50** | **q75** |
| B | 0.00 | 2.42 | 13.78 | 0.00 | 0.13 | 1.24 |
| C | 0.51 | 3.55 | 16.07 | 0.00 | 0.02 | 1.53 |
| D | 0.34 | 2.78 | 6.56 | 0.00 | 0.86 | 1.69 |
| E | 1.22 | 6.28 | 14.05 | 0.00 | 0.00 | 1.57 |

### 4.4 Wetness group assessment and changes

The eight *in situ* sample plots were grouped by their corresponding classification (pixel based) resulting in five sample plots for the 'Moist' and three sample plots for the 'Wet' CALU group (grouping B). Results confirmed the assumed differences in wetness between the groups (Figure A4). The sample plots assigned to the 'Wet' group were mostly dominated by *Equisetum* and *Poaceae* plants. One *in situ* sample plot contained more then 80% coverage of horsetail (*Equisetum*) and was classified as 'Wet', indicating high wetness. *Poaceae* was present in more then one sample plot. The 'Moist' group showed more heterogeneity in the vegetation data. The coverage of *Carex* and *Poaceae* reached from 0% to over 90%, covering a wide window on wetness information reaching from 'Moist' to 'Wet'.

Over a period of 10 years, the wetness inside the basins showed differed changes with respect to the bioclimatic subzones (see wetness groups in Table 1).

In subzone B, the 'Water' fraction decreased over seven years, while the 'Wet' group remained above 20% of the fraction for the DLBs in this subzone. The fraction of the 'Moist' group in the northern part of Yamal changed very slowly compared to the other subzones (see Figure 6), and the fraction of the 'Dry' group increased to over 50% after the fourth year, reaching more than 60% by the seventh year after the drainage event.

In subzone C, the 'Water' fraction experienced a decrease comparable to that observed in subzone B. The 'Wet' group showed a strong increase in the first five years after the drainage, followed by a gradual decrease. The 'Moist' group varied around 10 - 20 % and remained relatively stable over time. However, the 'Dry' group initially increased shortly after the drainage event, reaching up to 40 %, then decreased before increasing again.

The 'Water' fraction in subzone D remained relatively stable over time, consistently below 10%. The fraction of the 'Wet' group slowly decreased over the 10 year period, while the 'Moist' group showed a strong increase, reaching over 50%. In



contrast, the 'Dry' group fluctuated around 10%.

In subzone E, 'Water' slowly decreased over time. The 'Wet' group initially increased shortly after the drainage event, then decreased and fluctuated around 20 %. The fraction of the 'Moist' group showed a slow but steady increase, reaching over 50% fraction in the basin. *Vice versa*, the 'Dry' group rapidly decreased from over 20% after the first two years to around 10% and lower for the following eight years.

**Figure 6.** Fraction change of wetness groups (see Table 1) for different DLB ages (1 - 10) separated into the bioclimatic subzones (a - d).

### 4.4.1 NDVI compared to vegetation groups (group A)

different data points were available representing a basin at a certain age, used for the regression analyse. Grouping the data resulted in a reduction of data points, there were 267 data points for 'Water' (fraction maxed at 97%), 139 for 'Grassland' (max. 33%), 270 for 'Lichen and Moss' (max. 55%), 317 for 'Shrub Tundra' (max. 92%), 341 for 'Wetland' (max. 99%), 111 for 'Forest' (max. 12%) and 260 for 'Barren' (max. 93%). The fraction for the 'Forest' group never exceeded above 12 % and was therefore excluded. The correlation between the NDVI and the 'Shrub Tundra' (grouping 'A') was highest with $R^2 = 0.444$





(Figure 7). Negative correlation was detected for the 'Water' unit and the 'Barren' group whereas it is rather low in the latter case. The 'Wetland' and the 'Lichen and Moss' groups did not show a correlation with the NDVI ($R^2$ below 0.1).

**Figure 7.** NDVI compared to fraction for different vegetation types (a - e; see Table 1 for legend) and 'Water' (f). For each vegetation group the regression line is shown in black and the corresponding $R^2$. No outlier correction was applied.





## 4.4.2   NDVI compared to wetness gradients (group B)

The availability of data varied across the merged units from group B. There were 341 data points for 'Wet' (fraction maxed at 99%), 317 for 'Moist' (max. 93%) and 290 for 'Dry' (max. 99%). The highest correlation ($R^2 = 0.458$) was identified between the NDVI and the 'Moist' group. A negative correlation ($R^2 = -0.324$) for 'Water' (see Figure 7) and nearly no correlation ($R^2 = -0.039$) was detected between the NDVI and the 'Dry' and 'Wet' groups ($R^2 = 0.066$) (Figure 8).

**Figure 8.** (a - c) NDVI compared to DLB-fraction for different wetness gradients (see Table 1). For each wetness group the regression line is shown in black and the corresponding $R^2$ on the bottom right side. No outlier correction was applied.

## 5   Discussion

We showed that landcover changes in DLBs can not be detected with an analysis focusing on changes in the NDVI only. Additional information on succession stages and landscape change trajectories could be derived using the landcover units. For




example, change of 'Wetlands' or 'Lichen and Moss' dominated landcover are not represented in the changes of NDVI over time (see Figure 7 and 8). The NDVI values did not show any correlation with the CALU group 'Dry' ($R^2$ below 0.1) implying

that an NDVI-only related survey on DLBs misses out important dryness-wetness information during the first 10 years after a drainage event took place.

The utility of the consideration of bioclimate subzones for landcover change analyses in DLBs is particularly evident for the wetness gradients 'Dry' and 'Moist'. A higher fraction for the wetness group 'Dry' occurs in the northern bioclimate subzones

(B and C) compared to the subzones (D and E) located in the south (Figure 6). The comparatively high fraction of (partially) barren area for all bioclimatic subzones and all age groups agrees with previous observations of non-vegetated areas in recently drained basins (Lantz, 2017). Also our results showed that 'Barren' areas account for a larger proportion of recently drained area in northern subzones. A higher fraction of 'Moist' conditions was found in the basins in the south of the peninsula (Figure 6). Loiko et al. (2020) concluded that the productivity of individual ecosystems can be limited by nutrient availability. We

hypothesize that lakes in the south might have higher nutrition rich lake sediments compared to the north. This influences the colonizing of the basin in addition to different climate variables. Our results underline that the vegetation colonizing the basins exhibited differences between the southern bioclimatic subzones (D and E) and the northern subzones (B and C) (Figure 3). This agrees with previous *in situ* studies which have shown strong dependence of vegetation establishment on local climate and other factors such as difference in local species abundance (Lantz, 2017; Loiko et al., 2020). The 'Dry', 'Moist' and 'Wet'

separation from our remote sensing results for DLBs is supported by the field vegetation data although data were limited. The wetness group separation can be justified through indicator plants. For example horsetail *Equisetum* occupies the ecotope with the lowest soil fertility in wet areas with sandy sediments. Moist ecotopes are occupied by sedge meadows (*Carex*) in the elevated parts as also reported by Loiko et al. (2020).

DLBs have previously found to be sites of wetland formation (Lantz et al., 2022), however they have been mostly described as wetlands in older basins on centennial or millennial timescales. Following drainage, landcover of the area has a tendency to become more heterogeneous over time (Figure 5). Our study showed higher than the average percentage of 'Shallow water' (#2) and permanent wetland area (#3) for DLBs of all age groups compared to the percentage of the same landcover unit in the respective bioclimatic subzones (Figure 5). Small wetlands formed after the drainage in the basin (Figure 4). Those might

be covered with a thick (exceeding 10 cm) moss-grass layer and are located in wet depressions (Loiko et al., 2020). Methane fluxes differ between lakes, wet ecotopes and upland tundra (e.g. Schneider et al. (2009); Matthews et al. (2020)). A change in flux patterns can be therefore expected after lake drainage but the overall magnitude remains to be quantified.

We observed a higher ratio for 'Dry to moist tundra, partially barren' (#15) and 'Moist tundra, abundant forbs, dwarf to tall shrubs' (#16), which indicates the successive colonization of the basin by vegetation during the following years. The presence

of 'Partially barren' (#19) indicates that the area is still disturbed and vegetation succession is still an ongoing process after 10 years of drainage for the northern subzones compared to the southern part of the peninsula where barren areas are less present in the basin. DLBs are known to have dry to aquatic tundra areas with the distributions of these different conditions





depending on a multitude of factors such as time passed since drainage, local climate and basin topography (Jones et al., 2022). In particular the high fraction of 'Dry to aquatic tundra, dwarf shrubs' (landcover unit #8) compared to the surrounding average

of that unit might indicate a recently DLB for the bioclimate subzones C, D and E within the first 10 years (Figure 5).

Drainage events frequently lead to only partial drainage, causing water areas to persist within the drained basin. Remaining 'Water' fraction in our study sites dropped below 1% after 10 years for all bioclimatic subzones with water area decreasing over time after drainage. Previously published work has shown a continued change of the open water fraction within DLBs

over time, but it might vary for different basin age stages confirming Magnússon et al. (2020). This trend may reverse in time. For older basins, an increase of open water fraction in the form of secondary lake development has been reported (Jorgenson and Shur, 2007; Jones et al., 2022).

The difference between DLBs and their surrounding area in landcover unit distribution is evident for all bioclimatic subzones

(Figure 5). This supports the generally held assumption that lake drainage introduces landcover heterogeneity into the landscape (Bartsch et al., 2023b). Raynolds et al. (2019) used Advanced Very High Resolution Radiometer (AVHRR) surface temperature data from 1982 to 2003 for determination of the mean July temperature values to derive the bioclimate subzones. Due to the changing climate those mean values are not true any longer (compare with Figure A1), this is supported by recently studies, for example showed Ermokhina et al. (2023) that the July isotherms of 6 °C are changing northwards. This may have an impact

on zone specific results and updated boundaries should be considered for future studies.

The dependence of surface composition in DLBs on time passed since drainage is a phenomenon which has been observed for permafrost regions across the Arctic (Jorgenson and Shur, 2007; Hinkel et al., 2003). Previous research focused on C-band SAR data to distinguish between wetness gradients (Widhalm et al., 2016). It was highlighted that data availability was a problem and that their approach works only when there is no scattering for example due to tree trunks. The use of HH polarization

is beneficial in this case, but of limited availability from Sentinel-1 across the terrestrial Arctic. In our landcover approach we combine two different satellite sensors, SAR and multi-spectral data to address this issue. The analysis presented here is limited to a time period when satellite data were available and focuses only on a specific region. Longer time series and extending the study area would allow for analysis stretching beyond the 10 year mark, leading to additional insights of landcover change trajectories within DLBs. Increasing the number of study sites would provide additional insights of these trajectories over a larger

number of diverse basins. The timing of a lake drainage event was determined through visual inspection of available satellite imagery. This can lead to inconsistencies and an automatic approach would need to be considered when expanding the study area on a circumpolar scale. Our approach focuses on DLBs on the Yamal peninsula which drained recently (between 1998 - 2021). Nitze et al. (2018) interpreted trends of indices for change from 1999-2014 and Bergstedt et al. (2021) identified DLBs on the North Slope of Alaska, but the question, when the lakes drained (abrupt change), remains. Geological records showed

that abrupt changes, in the Earth system can arise from slow changes in one component (Brovkin et al., 2021). Paleorecords would be needed to understand the full cycles of DLBs on longer time scales.



To constrain the analysis and to better enable surface change assessment, basins were carefully selected, omitting those which were located in active floodplains or drained due to suspected anthropogenic influence. To further study the complex dynamics of post drainage landcover development, basins of different characteristics, including those connected to floodplains and with partial drainage of less than 50%, should be considered.

## 6 Conclusions

Improving our knowledge for processes following lake drainage events in Arctic environments associated with permafrost conditions and surrounding terrain is crucial for climate change impact assessments. Understanding the landscape changes associated with drainage, as described quantitatively through changes in landcover units in this study, is needed for representing consequences of permafrost thaw lake change in land surface and landscape models. These models depend on precise quantitative data. It could be shown that the landcover succession of recently drained basins follows a certain sequence of landcover changes as the ecosystem transitions from a uniform to a heterogeneous landcover. It differs with location on Yamal. The bioclimatic subzones (north-south climatic gradient) play an important role for the succession progression. Not only can DLBs be seen as hot spots of greening, but also important additional information regarding the development of wetlands was shown. Lake drainage leads to the development of wet ecotopes what is reflected in the landcover units but not directly represented through conventional NDVI analyses. Differentiation considering wetness is of relevance for methane fluxes. A realistic representation of wetness gradients in landsurface models may allow for improved descriptions of associated changes in future fluxes. In addition to carbon cycle impacts, the corresponding change of biodiversity needs to be addressed since drainage is a common feature across the entire Arctic. The presented approach can facilitate such investigations. In summary, our results advance the understanding of the development for recently DLBs across a range of bioclimatic subzones, drying versus wetting and landcover succession. The corresponding change considering the different identified trajectories of landcover change for the carbon cycle and species abundance (flora and fauna) remains to be addressed.

*Competing interests.* The authors declare no competing interests.

*Acknowledgements.* This work was supported by the Q-Arctic project and has received funding under the European Research Council (Grant Agreement No 951288). The CHARTER project has received funding under the European Union's Horizon 2020 Research and Innovation Programme under Grant Agreement No 869471. Further support has been received through the program of the Ministry of Science and Higher Education of the Russian Federation "Terrestrial ecosystems of northwestern Siberia: assessment of the modern transformation of the communities", No. 122021000089-9 as well as ESA Permafrost_cci and AMPAC-Net.



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



**Appendix**

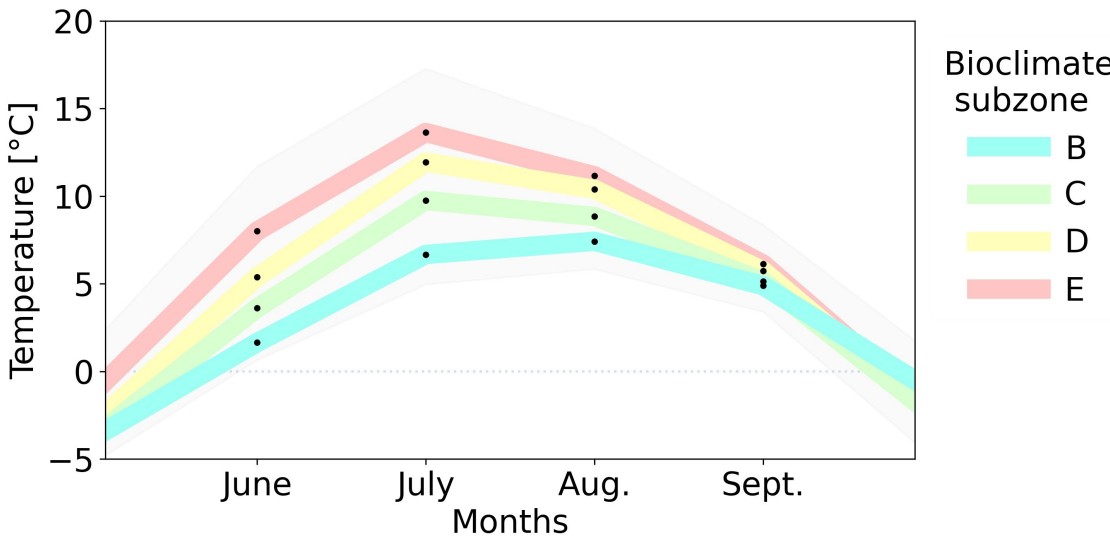

**Figure A1.** Mean temperature data from ERA5 for the Yamal peninsula separated by bioclimate subzones (see Figure 1). The covered data period is from 2015 to 2021. Only the months with a monthly mean higher than 0 °C are shown.

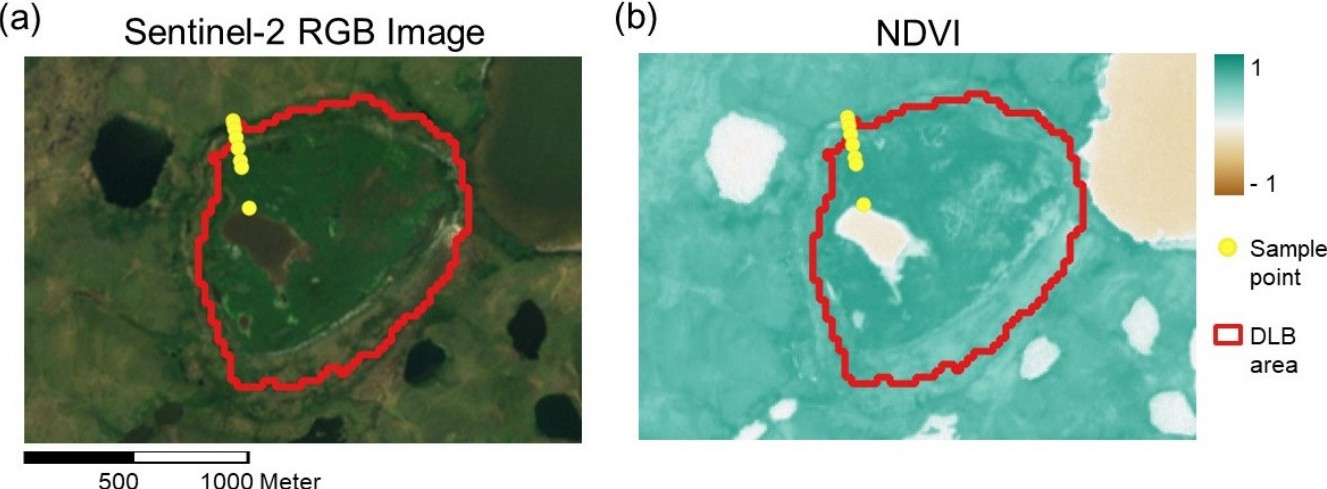

**Figure A2.** Overview of collected *in situ* data (2016) at a lake drained in 2010 (a) Red, green and blue (RGB) Sentinel-2 composite for the 20.07.2016, six days before the *in situ* data was collected. (b) corresponding Normalized Difference Vegetation Index (NDVI) values. For location see Figure 1 indicated as green triangle with the number 2, detailed coordinates are in Table A1.



**Table A1.** Drained lake basin *in situ* data, Yamal peninsula / Erkuta river (location 2 in Figure 1, Date: 26.07.2016. Start of transect P1: N68.18077° E69.05803°, end of the transect, P2: N68.17712° E69.05982°. Soil in old shore zone: clay, Soil on old lake bed: clay, vegetation cover on the remaining lake area 1 %.

|  | Plot 1 | Plot 2 | Plot 3 | Plot 4 | Plot 5 | Plot 6 | Plot 7 | Plot 8 |
|---|---|---|---|---|---|---|---|---|
| **Distance from 0 [m]** |  | 14 | 39 | 80 | 130 | 190 | 220 | 413 |
| **Bare soil cover [%]** | 0 | 0 | 0 | 0 | 0 | 0 | 2 | 5 |
| **Sand [%]** | 0 | 0 | 0 | 0 | 0 | 0 | 0 | 0 |
| **Clay [%]** | 0 | 0 | 0 | 0 | 0 | 0 | 2 | 5 |
| **Moss cover [%]** | 3 | 10 | 15 | 1 | 1 | 0 | 3 | 5 |
| **Carex [%]** | 77 | 1 | 50 | 30 | 1 | 0 | 0 | 0 |
| **Eriophorum [%]** | 0 | 0 | 0 | 50 | 85 | 0 | 0 | 0 |
| **Poaceae [%]** | 0 | 1 | 3 | 5 | 7 | 95 | 30 | 95 |
| **Forbs [%]** | 3 | <1 | 3 | 0 | <1 | 0 | 55 | 0 |
| **Salix [%]** | 0 | 0 | 0 | 0 | 0 | 0 | 0 | 0 |
| **Shrubs [%]** | 0 | 0 | 0 | 0 | 0 | 0 | 0 | 0 |
| **Equisetum [%]** | 0 | 80 | 20 | 0 | 0 | 0 | 0 | 0 |
| **Vegetation cover total [%]** | 80 | 90 | 90 | 85 | 95 | 95 | 98 | 95 |
| **Litter [%]** | 20 | 10 | 0 | 15 | 5 | 5 | 0 | 0 |

**Addition:**  - between Plot 4 and Plot 5 some *Salix* shrubs (about 7 shrubs), h≈50 cm

- Plot 5: soil under vegetation = sand

- between Plot 6 and Plot 8 vegetation like on Plot 6 with

*Eriophorum scheuchzeri* patches and *Carex* stans patches.





**Figure A3.** Vegetation survey plot 1, at the location 2 (Figure 1) inside the DLB. The lush vegetation differs from the surrounding area (Photo 26[th] July 2016, Dorothee Ehrich).



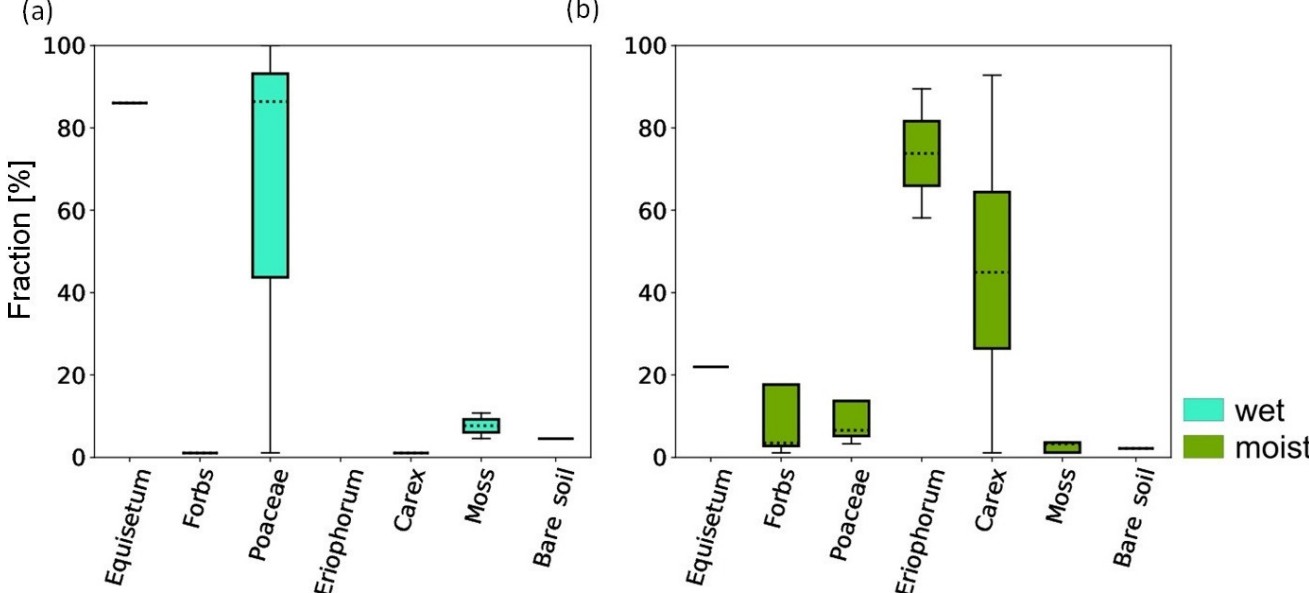

**Figure A4.** Types of surface data conducted in the field, separated for the corresponding classified pixel (a: Moist, b: Wet). Eight different surface categories were distinguished, vegetated (*Equisetum*, Forbs, *Poaceae*, *Eriophorum*, *Carex*, Moss) and none-vegetated (Bare soil).





**Figure A5.** Annual 'Water' fraction for different DLBs ages (1-10). The data were separated into the different bioclimate subzones.



**Table A2.** NDVI data from Figure 2.

| BCzone | age | min | q25 | q50 | q75 | max. |
|---|---|---|---|---|---|---|
| B | 0 | 0.244 | 0.299 | 0.339 | 0.440 | 0.636 |
| B | 1 | 0.174 | 0.203 | 0.256 | 0.395 | 0.832 |
| B | 2 | 0.178 | 0.23 | 0.339 | 0.406 | 0.804 |
| B | 3 | 0.160 | 0.276 | 0.327 | 0.362 | 0.838 |
| B | 4 | 0.219 | 0.236 | 0.283 | 0.49 | 0.822 |
| B | 5 | 0.152 | 0.233 | 0.261 | 0.310 | 0.920 |
| B | 6 | 0.119 | 0.180 | 0.200 | 0.585 | 0.658 |
| B | 7 | 0.276 | 0.458 | 0.518 | 0.519 | 0.519 |
| B | 8 | | | | | |
| B | 9 | | | | | |
| B | 10 | | | | | |
| C | 0 | -0.039 | 0.143 | 0.326 | 0.362 | 0.399 |
| C | 1 | 0.140 | 0.183 | 0.225 | 0.296 | 0.367 |
| C | 2 | 0.202 | 0.25 | 0.299 | 0.336 | 0.373 |
| C | 3 | 0.202 | 0.208 | 0.235 | 0.304 | 0.435 |
| C | 4 | 0.200 | 0.216 | 0.312 | 0.357 | 0.438 |
| C | 5 | 0.226 | 0.279 | 0.333 | 0.465 | 0.596 |
| C | 6 | 0.000 | 0.192 | 0.384 | 0.474 | 0.564 |
| C | 7 | 0.453 | 0.473 | 0.494 | 0.528 | 0.562 |
| C | 8 | 0.253 | 0.349 | 0.445 | 0.478 | 0.512 |
| C | 9 | 0.000 | 0.250 | 0.500 | 0.565 | 0.631 |
| C | 10 | 0.318 | 0.418 | 0.519 | 0.619 | 0.719 |
| D | 0 | -0.548 | -0.047 | 0.026 | 0.255 | 0.394 |
| D | 1 | -0.076 | 0.14 | 0.226 | 0.278 | 0.467 |
| D | 2 | 0.000 | 0.189 | 0.311 | 0.432 | 0.677 |
| D | 3 | 0.156 | 0.328 | 0.459 | 0.590 | 0.645 |
| D | 4 | 0.215 | 0.444 | 0.597 | 0.746 | 0.762 |
| D | 5 | 0.316 | 0.63 | 0.692 | 0.749 | 0.816 |
| D | 6 | 0.370 | 0.415 | 0.449 | 0.637 | 0.783 |
| D | 7 | 0.430 | 0.446 | 0.606 | 0.785 | 0.860 |
| D | 8 | 0.499 | 0.534 | 0.554 | 0.582 | 0.645 |
| D | 9 | 0.470 | 0.499 | 0.527 | 0.592 | 0.657 |
| D | 10 | 0.528 | 0.541 | 0.554 | 0.588 | 0.622 |
| E | 0 | -0.211 | -0.051 | 0.101 | 0.197 | 0.342 |
| E | 1 | 0.101 | 0.165 | 0.189 | 0.267 | 0.303 |
| E | 2 | 0.114 | 0.190 | 0.296 | 0.378 | 0.648 |
| E | 3 | 0.176 | 0.374 | 0.533 | 0.665 | 0.719 |
| E | 4 | 0.246 | 0.478 | 0.603 | 0.610 | 0.777 |
| E | 5 | 0.378 | 0.493 | 0.629 | 0.710 | 0.794 |
| E | 6 | 0.533 | 0.54 | 0.636 | 0.746 | 0.791 |
| E | 7 | 0.585 | 0.596 | 0.607 | 0.618 | 0.629 |
| E | 8 | 0.538 | 0.58 | 0.622 | 0.704 | 0.786 |
| E | 9 | 0.534 | 0.553 | 0.602 | 0.674 | 0.762 |
| E | 10 | 0.530 | 0.595 | 0.670 | 0.728 | 0.739 |