# Peer review of "Landcover succession for recently drained lakes in permafrost on the Yamal peninsula, Western Siberia"

_EGUsphere, 2024_

## Author Response (AR1)

Dear editor,

We have revised the manuscript as requested, carefully addressing and incorporating all the reviewers' comments and suggestions. We hope all suggested grammar and spelling changes have been implemented, and that we have addressed each comment.

A Supplement file is now added to the manuscript**. Table 4**. ('Water' fraction), **Figure A2**. (Overview of collected in situ data), **Figure A5**. (Annual 'Water' fraction) and **Table A2**. (NDVI data) have been moved there, along with the new table recommended by Reviewer #1.

Changed Figures and Tables:

| Old manuscript | Revised manuscript | Action |
|---|---|---|
| Table 4. | Table S1. | *moved* |
| Figure A2. | Figure S1. | *moved* |
| Figure A5. | Figure S2. | *moved* |
| Table A2. | Table S2. | *moved* |
| | Table S3. | *new* |

We would like to thank both reviewers for their very helpful and detailed revisions and comments. Below is our detailed response to each reviewer comment:

**Reviewer 1**

This study uses a land cover classification developed from Sentinel data to look at the vegetation of 51 drained thaw lake basins on the Yamal peninsula, and how land cover changes over the first years (up to 10) after drainage. All in all, the paper presents the methods and results adequately, but does not, either the Introduction or Discussion, convince the reader that this is important, interesting work. For example, the Discussion starts with the sentence, "We showed that landcover changes in DLBs can not be detected with an analysis focusing on changes in the NDVI only." Your most important result is not this negative finding, but rather that the landcover classes show the changing succession of land cover and consistent patterns in proportion of moisture groups during the years after lake drainage.

- Reply: Thank you for your valuable comment. Given that our study is exclusively focused on the Yamal Peninsula, we aimed to present the results with most caution. However, we rephrased:
  - Old: We showed that landcover changes in DLBs can not be detected with an analysis focusing on changes in the NDVI only. Additional information on succession stages and landscape change trajectories could be derived using the landcover units.
  - New: We showed the changing succession of land cover and consistent patterns in proportion of moisture groups during the years after lake drainage. In addition, our results convincingly show the capability of this landcover mapping approach to capture relevant post-drainage landcover change processes on a basin scale. Furthermore, we demonstrated that this change in the landcover cannot be detected with an analysis focusing solely on changes in NDVI.

The introduction does not convey the importance of drained thaw lake basins on the Yamal peninsula. There is mention of general permafrost thaw leading to $CO_2$ and methane emissions, but nothing directly related to thaw lake basins and their drainage. How do the permafrost characteristics and associated greenhouse gas emissions change with lake drainage? How are these lakes important

to wildlife or reindeer herders when they are full and when they are drained? Some of the information from the first paragraph of the study area section could be moved into the introduction, particularly lines 78-81.

- Reply: We added the following publications to our introduction and rephrased it:
  o Publications:
    ▪ Laptander, Roza / Horstkotte, Tim / Habeck, Joachim Otto / Rasmus, Sirpa / Komu, Teresa / Matthes, Heidrun / Tømmervik, Hans / Istomin, Kirill / Eronen, Jussi T. / Forbes, Bruce C. Critical seasonal conditions in the reindeer-herding year: A synopsis of factors and events in Fennoscandia and northwestern Russia 2024-03 Polar Science, Vol. 39 Elsevier BV p. 101016, https://doi.org/10.1016/j.polar.2023.101016
    ▪ Treat, Claire C. / Virkkala, Anna-Maria / Burke, Eleanor / Bruhwiler, Lori / Chatterjee, Abhishek / Fisher, Joshua B. / Hashemi, Josh / Parmentier, Frans-Jan W. / Rogers, Brendan M. / Westermann, Sebastian / Watts, Jennifer D. / Blanc-Betes, Elena / Fuchs, Matthias / Kruse, Stefan / Malhotra, Avni / Miner, Kimberley / Strauss, Jens / Armstrong, Amanda / Epstein, Howard E. / Gay, Bradley / Goeckede, Mathias / Kalhori, Aram / Kou, Dan / Miller, Charles E. / Natali, Susan M. / Oh, Youmi / Shakil, Sarah / Sonnentag, Oliver / Varner, Ruth K. / Zolkos, Scott / Schuur, Edward A. G. / Hugelius, Gustaf Permafrost Carbon: Progress on Understanding Stocks and Fluxes Across Northern Terrestrial Ecosystems 2024-02 Journal of Geophysical Research: Biogeosciences, Vol. 129, No. 3 American Geophysical Union (AGU), https://doi.org/10.1029/2023JG00763
  o New: (lines 78-81 moved to line 39) …et al., 2021). Disappearing lakes were previously reported specifically for the southern part of the Yamal Peninsula (Smith et al., 2005; Nitze, 2018). Central Yamal is known for rising temperatures and changes associated with unusually warm summers (thaw slumps, active layer deepening etc.; e.g. Babkina et al., 2019; Bartsch et al., 2019a). The whole region has been shown to be a hot spot of thaw lake change (Nitze, 2018).
  o New: (at line 27) …flora and fauna. The lakes are crucial for wildlife and reindeer herders both when full and drained (Kumpula et al., 2011; Laptander et al., 2024). When full, they provide essential water sources and support diverse ecosystems (Laptander et al., 2024). However, late freezing of lakes can delay reindeer migration, causing herders to move reindeer to winter pastures later than in previous decades (Kumpula et al., 2012; Laptander et al., 2024). Early snowmelt and unsafe ice cover can further complicate migration and increase the risk of avalanches and floods. When drained, the lakes create new grazing areas with abundant and diverse vegetation, attracting reindeer in the summer. This vegetation succession offers rich pastures but may pose health risks to reindeer due to unfamiliar plant species (Laptander et al., 2024).
  o New: (at line 43) …et al., 2022). Permafrost thaw often increases soil moisture and lake extent, generally enhancing $CH_4$ emissions (Schuur et al., 2022; Treat et al. 2024). In contrast, lake drainage significantly alters permafrost and greenhouse gas emissions by reducing soil moisture and changing hydrology, leading to decreased $CH_4$ emissions (Treat et al. 2024). However, predicting $CO_2$ and $CH_4$ fluxes is complex due to spatial variability in vegetation, soil carbon stocks, and geomorphology across the permafrost domain. Additionally, distinguishing between wetlands and lakes in

remote sensing data remains challenging, risking double counting of emission sources (Treat et al. 2024).

Section 4.1 – include, at least in the Appendix, a table which lists of all the DLBs included in the analysis, their drainage dates, area, and bioclimate subzone. It's not clear if all 51 lakes identified are include in the study, or if older ones may have been excluded (see comment below re. line 249-250).

- Reply: We included in the supplement file the following information for each DLB: ID, lat, lon, area, drainage year and subzone.

The inclusion of the peripheral area around the DLBs in the NDVI analysis is helpful. It shows that the method is capturing the relevant changes.

- Thank you.

In Section 4.3, Table 3 presents data that is shown in Figure 5. I suggest moving the table to the Appendix, as I see no benefit in making readers look through both the table and the figure.

- Reply: We moved Table 3 to the Appendix.

In Section 4.4, you report ground data for moisture class grouping (B), and describe cover for broad species groups. You do not compare your ground data with land cover units or land cover groups (A). You should report this and discuss these results, as you do the moisture class grouping. Also, refer to Figure A5 in this section (lines 311?).

- Reply: Given that the landcover product isn't designed for the purpose of distinguishing between different species, we found it beneficial to compare them based on their wetness grouping. Generally, increasing the amount of in situ data would enhance the precision of these comparisons. Additionally, the sample plots are in a line with varying intervals, rather than being equally distributed across the basin as it would be useful for such analysis. However, we included in the Table A1 the corresponding Landcover Unit and added this paragraph into the Discussion part:
  - (lines 359-361) …local species abundance (Lantz, 2017; Loiko et al., 2020). The landcover product is not designed to differentiate between various plant species. The grouping into the wetness gradients 'Dry', 'Moist', and 'Wet' is supported by field vegetation data, despite limitations. Enhanced precision in these comparisons could be achieved through increased in situ data availability. Additionally, distributing sample plots evenly across the basin, rather than along a line with varying intervals, would improve the analysis. However, the wetness separation is further justified by indicator plants (see Figure A4 and Table A1). For example…

Discussion – If the fact that there is more water retained in lakes in southern subzones is important enough to mention in the abstract (lines 12-13), then it should be discussed in the Discussion section.

- Reply: When extending the drainage data beyond the 10 years of drainage data (which we shortened due to the limited number of available basins), it appears that water levels for individual basins can fluctuate. Consequently, the remaining water decreases over the course of those 10 years, but various factors may influence further changes. Due to the lack of comprehensive data, we are careful in interpreting water fractions only. We added the following sentence into the Discussion:
  - (line 384) …over time after drainage. Our results show that more water retained in lakes after a drainage event in southern subzones for the first 10 years after drainage, however this can be biased since additional factors may also contribute to further

changes like floodplain related flooding or basin depth. Due to the lack of comprehensive data, we approach the interpretation of water fractions cautiously. Previously published work…

Line 6 – change "A regression analyses" to "Regression analyses"

- Revised as requested.

Lines 21-22 – rewrite sentence. Not clear what you are trying to say here.

- Reply: We rephrased the sentence to:
  o Old: The forming of characteristic landforms can be described as a process of the disturbance of the thermal equilibrium of the ground (Jones et al., 2011).
  o New: Temperature changes are responsible for disrupting the balance of heat in the ground and influence the formation of those characteristic landforms (Jones et al., 2011).

Line 24 – change "cycles span over thousands of years. Those" to "cycles can span thousands of years. These"

- Revised as requested.

Line 25 – introduce the acronym DLB

- Revised as requested.

Lines 44-45 – change "identified the largest changes within the first five years after drainage for the NDVI." to "identified the largest changes in NDVI within the first five years after drainage."

- Revised as requested.

Lines 63-67 – Most of this paragraph belongs in the methods. Maybe keep the second sentence, in the same paragraph as Lines 61-62.

- Reply: Part of this paragraph aims to clarify the objectives of the paper, so we have decided to keep the first part in the introduction. The second part has been moved to the beginning of the methods section.

Line 73 – change "Siberian Plain, the highest" to "Siberian Plain. The highest"

- Revised as requested.

Line 74 – change "is increasing" to "increases"

- Revised as requested.

Line 85-86 – move "the Vaskiny Dachi research station is located" to the beginning of the sentence

- Reply: In addition to reviewer #2 we rephrased the sentence to: The Vaskiny Dachi research station is situated in central Yamal, in bioclimatic subzone D, and has been monitoring extensive slope processes, including cryogenic landslides, since 1989 (Leibman et al., 2015).

Figure 1 – in inset, remove red dots within the Yamal box.

- Revised as requested.

Line 124 – change "is derived" to "was derived"

- Revised as requested.

Line 125 – change "is generated" to "was generated"

- Revised as requested.

Line 128 – change "is taken" to "was taken"

- Revised as requested.

Line 131 – change "achieve" to "archive"

- Revised as requested.

Line 131 – change "extents" to "extends"

- Revised as requested.

Line 133 – change "is used for our study. Data are available" to "was used for our study. Data were available"

- Revised as requested.

Line 142 – change "corresponding plant" to "corresponding vascular plant"

- Revised as requested.

Line 143 – change "50 species" to "50 vascular species"

- Revised as requested.

Line 146-154 – change "mean July temperatures" to "historic mean July temperatures". Your figure A1 shows that these temperatures are now higher

- Revised as requested.

Line 149 – change "and woody plants increase in stature to hemiprostrate dwarf shrubs (<15 cm tall)," to "and hemiprostrate dwarf shrubs (<15 cm tall) are present,"

- Revised as requested.

Line 159 – change "In a second step cloud free" to "In a second step, cloud free"

- Revised as requested.

Line 161 – change "Landcover was derived using Sentinel-1 and Sentinel-2 data " to "Landcover was modified from Bartsch et al. (2023a)" Or something similar indicating that the landcover built on this previous study.

- Revised as requested.

Line 187- change "temperature below freezing point was used only" to "only data with temperatures below the freezing point were used."

- Revised as requested.

Line 231 – change "has been" to "was"

- Revised as requested.

Line 242 – change "extend" to "extent"

- Revised as requested.

Line 244 – change "expands over" to "covers"

- Revised as requested.

Line 248-249 – how could a DLB age be -5? Does that mean it refilled 5 years ago? Please make this clear in the text.

- Reply: Our initial annual Landcover database covers basin ages from 5 years before the drainage event to 24 years after the event. In this database, the year of the drainage event is represented as year 0. We rephrased:
  - Old: Basin ages in our database ranged from -5 to 24 years, where year 0 represents the year of the event of the drainage.
  - New: Our annual Landcover database included Basin ages from 5 years before the drainage event to 24 years after the drainage event took place. The year 0 represents the year of the event of the drainage.

Lines 249-250 – please explain further why you excluded the DLBs that drained earlier than 2012. "age gaps and inconsistent time series respectively" is not clear. With respect to what? "Further analyses were therefore limited to a basin age of 10 years" does this mean that you excluded DLBs that drained more than 10 years ago or that you included only the first 10 years for lakes that had drained more than 10 years ago?

- Reply: The issue arises when we have older basins and aim to apply a fully consistent space-for-time approach. To avoid the gaps in the space-for-time sequence, we've set a limit on basin age to 10 years. This ensures that we maintain a consistent approach and have a relatively adequate number of basins covering various drainage dates. Because we distinguish between the Bioclimate Subzones, we're unable to allocate full 10 years to each Subzone. Consequently, in Figure 5, we increased the number of available basins by combining the different ages into two groups (1 - 5 and 6 - 10 years). We rephrased:
  - Old: Splitting the data into the bioclimate subzones resulted in age gaps and inconsistent time series respectively. Further analyses were therefore limited to a basin age of 10 years. Age groups (1 - 5 and 6 - 10 years) were built to increase the number of recently DLBs.
  - New: The separation of the data into the Bioclimate Subzones resulted in data gaps for the more northern Subzones and no full data coverage of 10 years could be achieved. Further analyses (see Figure 5) were therefore carried out introducing two age groups (1 - 5 and 6 - 10 years).

Lines 256-257 – Please explain either here or in the discussion why the non-drained lakes in subzones B&C had such high NDVI values. Are they generally shallower with emergent vegetation?

- Reply: Thank you for bringing up this interesting aspect. We hadn't considered the depth of the lakes yet. We suggest including the following sentence:
  - (At line 257) The NDVI values in the Arctic region increase from colder to warmer bioclimate subzones, which is also shown in Figure 2.

Lines 264-265 – this small percent change is not worth mentioning, since it is based on just one lake. The general patterns of decreasing water and wetland types, and increasing barren and dry types is the main result. Lines 266-271 – Way too much detail for such a small sample size. Please report more general patterns, such as that #15 dry to moist tundra, partially barren was an important component of all these DLB, at all ages, but not in Subzone B. Lines 272-276 – de-emphasize changes in years 9 and 10, when you only have 3 lakes.

- We rephrased the paragraph.

Figure 3 – The brown color for #19 partially barren is different in the top two vs. bottom two bar charts. They should be changed to match. I prefer the top, which has a stronger contrast between #8 and #19 (and for Figure 4 also). Since you include color boxes for the subzones, in the caption you should refer readers to Figure 1 for Subzone mapping.

- We checked the color code, it is the same.

Line 283 – change "'Water'" to "#1 'Water'"

- Revised as requested.

Line 284 – change "'Wetlands'" to "#2 'Wetlands'" – be consistent and always include the unit # (on Line 286 as wel)l. It is difficult for the reader to have to constantly refer back to Table 1.

- Revised as requested.

Line 288 – remove "CAL unit" for simplicity and consistency.

- Revised as requested.

Figure 4 caption – include the information that this lake is located at site 1 in Figure 1.

- Revised as requested.

Line 289 – I would add that this progression for the first 6 years at one lake matches well with the patterns recorded for the set of Subzone E DLBs.

- Revised as requested.

Line 295 – "higher proportion" – higher than what? Ahah – than the average overall cover in that subzone. Add reference to Figure 4.

- We added following sentence in the caption: The lake extend is derived from (Nitze et al., 2018.

Lines 297-301 (including Table 4) – move to Supplement – not particularly meaningful. Decrease in water cover expected, and shown by subzone in figures 3 and 5

- moved to Supplement.

Line 303 – change " 'Wet' CALU group (grouping B)." to " 'Wet' group (grouping B, Table 1)."

- Revised as requested.

Line 307-308 – explain this more fully: "a wide window on wetness information reaching from 'Moist' to 'Wet'."

- Reply: We changed the sentence (line 307):
  o Old: The coverage of Carex and Poaceae reached from 0% to over 90%, covering a wide window on wetness information reaching from 'Moist' to 'Wet'.
  o New: A broad range of fractional cover, spanning from 0% to over 90%, is observed for Poaceae in the wet group and for Carex in the moist group.

Lines 302-308 – You compare the ground data with wetness classes, so that we have some idea that the wetness classes matched the ground data. You give us no information on how the ground data match with the land cover units or the plant physiognomy (Group A) classes.

- Reply: see Comment 4). We included the corresponding Landcover Unit in Table A1. Due to the limited number of field data points available, we chose to compare them using a smaller set of distinct categories. From the 19 different Landcover Units, Group B includes only four (water, wet, moist, dry), whereas Group A includes seven categories (water, wetland, grassland, lichen and moss, shrub tundra, forest, and barren).

Lines 310-311 – need to introduce Figure 6

- Revised as requested.

Line 315 – deemphasize results based on n=1. You can say that the fraction of the dry group increased over time, staying above 50% after 4 years.

- Reply: We rephrased:
  - Old: … and the fraction of the `Dry' group increased over 50% after the fourth year, reaching more than 60% by the seventh year after the drainage event.
  - New: … and the fraction of the `Dry' group increased over time, staying above 50% after 4 years

Line 328 – change "analyse" to "analysis"

- Revised as requested.

Line 328 – "341 different data points were available representing a basin at a certain age" Please clarify. Do you mean that you averaged the NDVI values for each DLB, to get one average value for each lake for each time period?

- Reply: Yes, we averaged the NDVI values for each DLB for each age. We rephrased:
  - Old: 341 different data points were available representing a basin at a certain age, used for the regression analyse.
  - New: 341 different data points were available for the regression analyses, with each data point representing a specific basin at a certain age. These data points include the mean NDVI value for that basin at the corresponding age, along with the fraction of the Landcover Group (e.g., Barren). One basin can have multiple data points if there are data available for different ages of that basin, which may vary for other basins.

Line 332 – change "correlation between the NDVI and the 'Shrub Tundra'" to "correlation between the NDVI and the percent cover of 'Shrub Tundra'"

- Revised as requested.

Line 333 – "it is rather low" – what does "it" refer to – NDVI or correlation (which is negative) or the strength of the correlation (R value)

- Reply: thanks for spotting the mistake, we removed the minus (–) in the R square values in the Figures, and rephrased:
  - Old: Negative correlation was detected for the 'Water' unit and the 'Barren' group whereas it is rather low in the latter case.
  - New: Negative correlation was detected for the 'Water' unit and the 'Barren' group whereas the R value is rather low in the latter case.

Line 334 – and the "Grassland" group

- Revised as requested.

Line 344 – delete "CALU"

- Revised as requested.

Line 345 – delete "out"

- Revised as requested.

Line 394 – change "showed Ermokhina et al. (2023)" to "Ermokhina et al. (2023) showed"

- Revised as requested.

Line 409 – change "question, when" to "question of when"

- Revised as requested.

Line 410 – delete comma

- Revised as requested.

Line 417 – change "knowledge for processes" to "knowledge of processes"

- Revised as requested.

Line 421 – change "It could be shown" to "We showed"

- Revised as requested.

Line 425 – change – "ecotopes what is reflected" to "ecotopes, which is reflected"

- Revised as requested.

Figure A2 caption - Describe which dataset the red line defining the DLB came from

- Included the sentence "The lake extent in the red line is provided by the Landsat product from Nitze (2018)."

Table A1 – include the corresponding imagery information for each sample point – landcover unit and group (A and B) of the nearest pixel (or group of pixels)

- We included a new row with the corresponding landcover unit.

Figure A4 caption – change "conducted" to "collected"

- Revised as requested.

**Reviewer 2**

The article "Landcover succession for recently drained lakes in permafrost on the Yamal peninsula, Western Siberia" investigates the succession of landcover changes in drained lake basins (DLBs) on the Yamal Peninsula using a novel landcover unit retrieval scheme tailored for the Arctic tundra biome. This approach aims to enhance the understanding of landcover changes compared to the traditional Normalized Difference Vegetation Index (NDVI) trend analyses. The study contributes to a better understanding of DLB landcover changes in permafrost environments, with implications for geomorphological, hydrological, and ecological development in these regions. The article's goals are well addressed in the results and discussion sections. The results section clearly presents the findings related to landcover changes and their correlation with NDVI, while the discussion interprets these findings and their implications for permafrost regions.

**Comments**

**1. Introduction**

- **Page 1, Line 17-18:** "Arctic permafrost regions are undergoing drastic changes due to climate change, leading to widespread, unprecedented landscape disturbances and changes (IPCC, 2022)." - The authors use the word "change" freely in this sentence and it can get confusing. Consider revising to something like the following: "Arctic permafrost regions are undergoing drastic disturbances as a result of climate variability, leading to widespread, unprecedented landscape transformations (IPCC, 2022)."

    - Revised as requested.

-**Page 1, Line 18-19:** "Those are both of gradual and abrupt nature and range from plot scale to largescale, a regional to circumpolar phenomenon (Turetsky et al., 2020)." -The word "those" is very vague. Instead, please list 2-3 of the main arctic permafrost landscape disturbances. Something like the following: "Common arctic permafrost landscape transformations, such as retrogressive thaw slumps, coastal erosion, and drained lake basins are both of gradual and abrupt nature and range from plot scale to landscape scale, a regional to circumpolar phenomenon (Turetsky et al., 2020)."

    - Revised as requested.

-**Page 1, Line 19-21:** "Increasing temperatures induce the thawing of ice-rich permafrost or the melting of ice and greenhouse gases, methane and carbon dioxide are released into the atmosphere (Turet-sky et al., 2020; Manasypov et al., 2022; Schuur et al., 2022)." -It is difficult to interpret this sentence. Instead, consider something like the following: "Warming induces the thawing of ice-rich permafrost landscapes, triggering a release of critical greenhouse gases into the atmosphere, such as methane and carbon dioxide through decomposition (Turetsky et al., 2020; Manasypov et al., 2022; Schuur et al., 2022)."

    - Revised as requested.

-**Page 2, Line 21-22:** "The forming of characteristic landforms can be described as a process of the disturbance of the thermal equilibrium of the ground (Jones et al., 2011)." -I suggest you modify this sentence to build on the previous one. Maybe something like the following: "Additionally, warming permafrost triggers the formation of characteristic landforms (i.e. polygon-shaped features) and can be defined as a process of the disturbance of the thermal equilibrium of the ground (Jones et al., 2011)."

    - Revised as requested.

-**Page 2, Line 22-24:** Please join sentences: "Closed depressions filled with water, formed by the settlement of the ground caused by the thermo-induced process, are called thermokarst lakes and their formation and drainage cycles span over thousands of years."

- Revised as requested.

-**Page 2, Line 31-32:** I would suggest removing references older than 10 years (Smith, Carroll and Kanevskiy) as they no longer can be considered "recent".

- We removed „recent".

-**Page 2, Line 36-38:** "The detection of previously drained lakes requires spatial data analyses in order to identify relevant depressions and gain knowledge about trajectories of landsurface hydrology change and vegetation succession is needed." -Consider rewriting this sentence, "is needed: at the end seems out of place.

- We rephrased this sentence to: Detecting previously drained lakes requires spatial data analyses to identify relevant depressions and gain knowledge about land surface hydrology changes and vegetation succession trajectories.

-**Page 2, Line 38:** "The applicability of remote sensing data for identifying DLBs has been demonstrated (Smith et al., 2005; Jones et al., 2011; Bergstedt et al., 2021)." -Please consider removing this sentence. Seems out of place.

- We removed this sentence.

-**Page 2, Line 43-44:** "Previous studies have focused on the usage of the Normalized Difference Vegetation Index (NDVI)." -To do what? Authors can be more specific. Maybe something like this: "Previous studies have focused on the usage of the Normalized Difference Vegetation Index (NDVI), to detect changes across DLBs. This popular spectral index is calculated by using the ratios between the red and near-infrared parts of the electromagnetic spectrum and is commonly used as a proxy to detect vegetation productivity and change across the Arctic."

- Reply: Thank you for the comment, we will be more specific and suggest following rephrasing:
  - Old: Previous studies have focused on the usage of the Normalized Difference Vegetation Index (NDVI).
  - New: Chen et al. (2021) and Liu et al. (2023) have used the Normalized Difference Vegetation Index (NDVI) to detect vegetation productivity and changes in DLBs. This popular spectral index is calculated by using the ratios between the red and near-infrared parts of the electromagnetic spectrum.

-**Page 2, Line 45-48:** "A spatial analysis of a multi-basin data set of landcover succession in recently drained basins has not been addressed yet. Mapping landcover and vegetation communities in the Arctic requires specialised approaches (Bartsch et al., 2016). Methodologies must be able to monitor the unique characteristics of Arctic landscapes and vegetation communities." -Are these included to suggest current limitations? These sentences seem out of place. Consider moving them to a different location (maybe have limitations as a separate paragraph?).

- Reply: These sentences are supposed to be the introduction of the landcover state of the art. The intention is to clarify why landcover has not been used until now for this purpose. We suggest the following rephrasing:
  - Old: A spatial analysis of a multi-basin data set of landcover succession in recently drained basins has not been addressed yet. Mapping landcover and vegetation

communities in the Arctic requires specialised approaches (Bartsch et al., 2016). Methodologies must be able to monitor the unique characteristics of Arctic landscapes and vegetation communities.

- o New: An analysis of land cover succession in recently drained lake basins has not been addressed yet. One of the reasons is that the generation of land cover data requires a specialized approach tailored to the unique characteristics of Arctic landscapes and vegetation communities (Bartsch et al., 2016).

**2. Study Area and Data**

**2.1 The Yamal Peninsula**

- **Page 3, Line 70-71:** "The Yamal peninsula (Western Siberia) has been subject to a multitude of different environmental changes over the past and represents multiple bioclimatic subzones (Figure 1)." -Revise to "The Yamal Peninsula (Western Siberia) has undergone numerous environmental changes over the years and encompasses multiple bioclimatic subzones (Figure 1)."

    - Revised as requested.

- **Page 3, Line 85-87:** "In central Yamal, in bioclimatic subzone D, the Vaskiny Dachi research station is located and extensive slope process activation resulting in cryogenic landslides are monitored since 1989 (Leibman et al., 2015)." -Consider revising, not a complete sentence.

    - Rephrased the sentence to: The Vaskiny Dachi research station is situated in central Yamal, in bioclimatic subzone D, and has been monitoring extensive slope processes, including cryogenic landslides, since 1989 (Leibman et al., 2015).

- **Page 4, Figure 1:** Suggestions: -It is difficult to see the Yamal Peninsula from (a) alone. It might be best to exclude parts outside of Siberia on the map and focus more on areas surrounding the peninsula. It seems to me that the permafrost fraction is quite variable in this area, which would be nice to highlight but it is difficult to see when you include the entire Arctic region in the map. I would also change the order of (b) and (c) so that the higher resolution permafrost fraction map is next to the larger scale map, showing the same units.

    - Reply: We apologize for the confusion. Including the comment from Reviewer#1, we updated Figure 1 in the manuscript. Specifically, we removed the red dots within the Yamal box in inset (a), we adjusted the extent of inset (a) to show more detail of the surrounding areas, we clipped the extend from Finland in the west to Krasnoyarsk Krai in the east, and we changed the order of (b) and (c).

- **Page 4, Line 92:** "Monthly mean temperatures of the bioclimatic subzones reflect the gradient (Figure A1)." -What gradient? Do the authors mean figure 1(a) NOT figure A1? This is confusing.

    - Reply: Sorry for the confusion. We rephrased:
        - o Old: Monthly mean temperatures of the bioclimatic subzones reflect the gradient (Figure A1).
        - o New: The temperature gradient over the study area represented in the bioclimatic subzones is shown in Figure A1 as monthly means.

**2.2 Data**

- **Page 4, Line 96-98:** "The presented study builds on data from the Sentinel missions derived via the Europe's Copernicus programme run by the European Commission (EC) and the European Space Agency (ESA)." -Consider rephrasing for clarity: "This study utilizes data from the Sentinel missions,

part of Europe's Copernicus programme operated by the European Commission (EC) and the European Space Agency (ESA)."

- Revised as requested.

**2.2.2 Auxiliary Data**

- **Page 5, Line 116-117:** "Daily temperature data are required for Sentinel-1 scene selection. They are derived using the ERA5-Land dataset product from the Copernicus Climate Change Service (C3S) at the European Centre for Medium-Range Weather Forecasts (ECMWF)." -Consider merging into one sentence for conciseness: "Daily temperature data, required for Sentinel-1 scene selection, are derived from the ERA5-Land dataset by the Copernicus Climate Change Service (C3S) at the European Centre for Medium-Range Weather Forecasts (ECMWF)."

- Revised as requested.

**4. Results**

- **Page 10, Line 242:** "The open water extend (before drainage) ranged between 0.02 km2 and 6.88 km2. -Consider changing "extend" to "extent". Fix this common grammar error throughout manuscript. "The open water extend (before drainage) ranged between 0.02 km2 and 6.88 km2.

- Revised throughout the manuscript.

**5. General comments:**

I believe the discussion is missing comments on differences in the spatial resolutions between the in-situ (2x2m plots) vs. Sentinel's 10m.

- Reply: We added the following sentence in addition to our reply to Reviewer #1's comment:
  o Old Suggested change (revierwer#1): …Enhanced precision in these comparisons could be achieved through increased in situ data availability. Additionally, distributing sample plots evenly across the basin, rather than along a line with varying intervals, would improve the analysis. However, the wetness separation…
  o New: …Enhanced precision in these comparisons could be achieved through increased in situ data availability. Additionally, distributing sample plots evenly across the basin, rather than along a line with varying intervals, would improve the analysis. We suggest one plot on a 10x10 m scale to cover each pixel from the landcover data product. However, for our in situ data, the wetness separation…

It seems the authors only collected in situ data at one DLB but failed to comment on which bioclimatic zone that fell in. Is this representative of DLBs in different bioclimatic zones?

- Reply: Thank you for the comment. The location of the in situ data is indicated in Figure 1 by a green triangle with the number 2 on top. To clarify the corresponding subzone, we changed the following sentences:
  o Old: (Figure A2) ...at a lake drained…
  o New: (Figure A2) …at a lake located in subzone E drained…

  o Old: (Table A1) …Drained lake basin in situ data…
  o New: (Table A1) …Drained lake basin located in subzone E in situ data…

  o Old: (Figure A3) …Vegetation survey plot 1, at the location 2…
  o New: (Figure A3) …Vegetation survey located in subzone E, plot 1, at the location 2…

- o Old: (Figure A4) …Types of surface data conducted in the field…
- o New: (Figure A4) …Types of surface data collected in the field in subzone E…

I suggest the authors revise parts of the manuscript to help it flow better, particularly the introduction, study area and methods. I believe many sentences are short and can be joined with those that follow to improve clarity and flow (example page 2, line 45-48).

- Reply: Thank you for your feedback. We believe that fusing the suggested changes, along with the comments from both reviewers, improved the flow of the manuscript. Additionally, we combined and reviewed other short sentences.

The abstract is well written, however it is misleading. The authors suggest the presented approach is better than NDVI, but they use NDVI to inform the space for time analysis, so how is that better? I suggest you reword the abstract and specifically mention that even though NDVI alone might not be best to capture changes in DLBs, it is still useful data that can help inform researchers and/or models for improved understanding.

- Reply: Thank you for raising this point. Our approach is intended to complement NDVI rather than replace it. While NDVI has numerous advantages (such as ease of derivation), it also has limitations when used exclusively for monitoring Drained Lake Basins as we have discussed in our study. However, we appreciate your feedback and changed these two sentences from our abstract:
  - o Old: The added value compared to commonly used Normalized Difference Vegetation Index (NDVI) trend analyses is demonstrated.
  - o New: The added value and complementarity of landcover units and the commonly used Normalized Difference Vegetation Index (NDVI) trend analyses is shown.

  - o Old: There was no correlation (R2 = 0.066) found between NDVI and the fraction of group 'Wet'. This highlights the importance of an alternative to NDVI such as the use of landcover units to describe wetland area changes.
  - o New: There was no correlation (R2 = 0.066) between NDVI and the fraction of the 'Wet' group. This underlines the need to complement NDVI analyses, such as the use of landcover units to describe changes in wetland areas.

Figure 2, 3: the caption reads subzone (a-d) but only figures for b-d are shown.

- Reply: Thank you for bringing up this issue. Bioclimate subzones B, C, D and E are present in our study area and the captions refer to the subplots (a-d) representing each of these subzones. We tried to address this by referencing subplots using lower case letters and using capital letters for the bioclimate subzones. However, rephrased:
  - o Old: Figure 2. NDVI change within the first ten years after drainage separated by bioclimatic subzone (a - d).
  - o New: Figure 2. Subplots (a - d), NDVI change within the first ten years after drainage separated by bioclimatic subzone (B – E).

I would like to see the landcover unit legends next figures 3, 4 and 5. It is cumbersome having to scroll back and forth to look at colors to match between table and map. Would make it easier for the reader to make links.

- Reply: Thank you for raising this issue. If we stick to the current page layout and add the legend, the data plots would become smaller due to limited page space. One potential

solution is to use landscape pages, which offer more space. However, we believe the editor should decide on the best approach to address this issue.

---

## Editor Decision (ED1)

[revised manuscript text omitted]
 (6-10)** | 0.00 | 3.03 | **9.03** | **8.80** | 1.06 | 0.71 | 0.59 | 3.99 | 5.93 | **9.18** | 2.91 | 0.21 | 0.06 | 0.87 | **5.62** | **6.78** | 0.00 | 0.49 | 0.56 |

---

## Author Response (AR2)

Detailed response:

We have revised the manuscript and addressed the editor's comments for the minor revisions.

**1)** Lines 3-4: Please rewrite this sentence.
- Reply: Thanks for the comment, we rephrased the sentence
    - Old: The added value and complementarity of landcover units and the commonly used Normalized Difference Vegetation Index (NDVI) trend analyses is shown.
    - New: The complementarity between landcover units and NDVI analyses is shown.

**2)** Line 10: Do you mean the inconsistency in the relationship? Can you please specify? **3)** And Can you please indicate with what?
- Reply: yes, that's what we mean, we rephrased it to:
    - Old: This underlines the need to complement NDVI analyses, such as the use of landcover units to describe changes in wetland areas.
    - New: The inconsistency in the association between those variables underlines the need to complement NDVI analyses with a scheme representing wetness, such as the use of landcover units to describe changes in wetland areas.

**4)** Introduction: The first part of the introduction is a very long paragraph. Can you please break this into three to four parts/paragraphs to help the readers?
- Reply: The introduction is now divided into several shorter paragraphs to enhance readability.

**5)** Line 76: You have done a good job explaining what are available methods up to date. But before going into the aim of the study, can you please add a very brief description of what is lacking in the current state of the art in this field?
- Reply: Thanks for the input, we included following sentence before "The aim of this study…":
    - New: Overall, NDVI analyses have been shown to provide valuable information on vegetation recovery after lake drainage. However, an approach representing changes in wetness is lacking. New landcover description schemes using recent satellite observations can provide relevant information and may thus complement NDVI analyses.

**6)** Table 1: Please add some more description about the color schemes so that the legend is standalone and the readers don't have to go back to Bartsch et al. 2024 to find them.
- Reply: We changed the caption of Table 1 and renamed the ID column to new ID:
    - Old: Table 1. Legend for the land cover units based on Bartsch et al. (2024) and grouping schemes.
    - New: Table 1. Legend for the landcover units based on Bartsch et al. (2024) and grouping schemes. The original IDs 18, 19 and 20 (representing different forest types) were merged into the new ID 18. The original ID 21 is now ID 19.

**7)** Figure 2: Please indicate that the subzone and the color bars next to the subzone are to be referred to from Table 1.
- Reply: Thank you for the comment. We changed the captions:
    - Old: Figure 2. Subplots (a - d), NDVI change within the first ten years after drainage separated by bioclimatic subzone (B - E). The peripheral area refers to the area around the DLBs for a zone of 1 km.
    - New: Figure 2. Subplots (a - d), NDVI change (Sentinel-2) within the first ten years after drainage separated by bioclimatic subzone (B - E). For Bioclimate subzone location and color legend see Figure 1. The peripheral area refers to the area around the DLBs for a zone of 1 km.

- o Old: Figure 3. Landcover change in time for the DLBs separated by the different bioclimate subzones (a - d). The sum of available basins for a certain age is displayed on top. For color legend see Table 1.
- o New: Figure 3. Subplots (a - d), Landcover change in time for the DLBs separated by the different bioclimate subzones (B - E). The sum of available basins for a certain age is displayed on top. For Landcover color legend see Table 1, for Bioclimate subzone location and color legend see Figure 1.

- o Old: Figure 6. Fraction change of wetness groups (see Table 1) for different DLB ages (1 - 10) separated into the bioclimatic subzones (a - d).
- o New: Figure 6. Subplots (a - d), Fraction change of wetness groups (see Table 1) for different DLB ages (1 - 10) separated into the bioclimatic subzones (B - E). For Bioclimate subzone location and color legend see Figure 1.

**8)** We also included the "Author contribution" section:
- o New: AB developed the initial concept which was adapted by CvB. CvB processed the data, analyzed the results and wrote the first draft of the manuscript. AB, HB, BW, AE, TK, and DE contributed to the conception of the study and writing of the manuscript, AS and SA provided additional feedback. In-situ data was collected by DE, AS and SA.